



# Iodine oxoacids and their roles in sub-3 nanometer particle growth in polluted urban environments

Ying Zhang[1,2,3,★], Duzitian Li[2,3,★], Xu-Cheng He[4,5], Wei Nie[2,3], Chenjuan Deng[6],

Runlong Cai[4], Yuliang Liu[2,3], Yishuo Guo[1], Chong Liu[2,3], Yiran Li[6], Liangduo Chen[2,3],

Yuanyuan Li[2,3], Chenjie Hua[1], Tingyu Liu[1], Zongcheng Wang[1], Lei Wang[2,3], Tuukka Petäjä[4], Federico Bianchi[4], Ximeng Qi[2,3], Xuguang Chi[2,3], Pauli Paasonen[4], Yongchun Liu[1], Chao Yan[2,3], Jingkun Jiang[6], Aijun Ding[2,3], Markku Kulmala[1,2,3,4]

[1]Aerosol and Haze Laboratory, Beijing Advanced Innovation Center for Soft Matter Science and Engineering, Beijing University of Chemical Technology, Beijing, China

[2]Joint International Research Laboratory of Atmospheric and Earth System Sciences, School of Atmospheric Sciences, Nanjing University, Nanjing, China

[3]Jiangsu Provincial Collaborative Innovation Center of Climate Change, Nanjing, China

[4]Institute for Atmospheric and Earth System/Physics, Faculty of Science, University of Helsinki, Helsinki, Finland

[5]Finnish Meteorological Institute, Helsinki, Finland

[6]State Key Joint Laboratory of Environment Simulation and Pollution Control, State Environmental Protection Key Laboratory of Sources and Control of Air Pollution Complex, School of Environment, Tsinghua University, Beijing, China

[★]These authors contributed equally to this work.

*Correspondence to*: Xu-Cheng He (xucheng.he@helsinki.fi) and Wei Nie (niewei@nju.edu.cn)

**Abstract.** New particle formation processes contribute significantly to the number concentration of

ultrafine particles (UFP), and have great impacts on human health and global climate. Iodine oxoacids ($HIO_x$, including iodic acid, $HIO_3$ and iodous acid, $HIO_2$) have been observed in pristine regions and proved to dominate NPF events at some sites. However, the knowledge of $HIO_x$ in polluted urban areas is rather limited. Here, we conducted a long-term comprehensive observation of gaseous iodine oxoacids and sulfuric acid in Beijing from January 2019 to October 2021 and also in Nanjing from March 2019 to

February 2020, and investigated the contribution of $HIO_x$ to UFP number concentration in urban environments. $HIO_3$ concentration is highest in summer, up to $2.85 \times 10^6$ cm$^{-3}$ and $2.78 \times 10^6$ cm$^{-3}$ in Beijing and Nanjing, respectively, and is lowest in winter, with a more prominent seasonal variation than $H_2SO_4$. $HIO_3$ concentration shows a clear diurnal pattern at both sites with a daily maximum at around



noontime, similar to the atmospheric temperature, radiation and ozone ($O_3$) levels. $HIO_2$ concentration

has the same diurnal and seasonal trend as $HIO_3$ but is overall about one order of magnitude lower than

$HIO_3$ concentration. Back trajectory analysis suggests that the sources for inland iodine species could be

a mix of marine and terrestrial origins, both having peak iodine emission in warm seasons. While the

contribution of $HIO_2$ to particle growth is marginal in Beijing and Nanjing, our results demonstrate that

$HIO_3$ enhances the particle survival probability of sub-3 nm particles by about 40% (median) and

occasionally by more than 100% in NPF events, suggesting $HIO_x$ are non-negligible contributor to UFPs

in polluted urban areas. As the growth contribution from $HIO_3$ and $H_2SO_4$ is similar on a per-molecule

basis, we propose that the sum of $HIO_3$ and $H_2SO_4$ could be used to estimate sub-3 nm particle growth

of inorganic acid origin, in the polluted atmospheres with a significant amount of $HIO_x$.

## 1 Introduction

Aerosol particles are ubiquitous in Earth's atmosphere and have both primary and secondary sources

(Kulmala et al., 2004b). Primary aerosol emissions include natural sources including the emission of sea

spray, release of soil mineral dust, emission of biomass burning smoke, and the injection of volcanic

debris (Claudio Tomasi, 2017) and anthropogenic emissions such as fuel combustion, industrial

processes and transportation (Claudio Tomasi, 2017). Besides direct emissions, atmospheric new particle

formation (NPF), a secondary particle source, plays a significant role in increasing aerosol population

(Kulmala et al., 2012). Only a few vapours, such as sulfuric acid ($H_2SO_4$), water vapour ($H_2O$), ammonia

($NH_3$), amines (e.g., dimethylamine, $C_2H_7N$) and highly oxygenated organic molecules (HOMs), are

widely confirmed to nucleate and form new particles under atmospheric conditions (Kulmala et al.,

2004a; Kürten et al., 2016; Li et al., 2020; Yao et al., 2018; Almeida et al., 2013; Kirkby et al., 2011;

Kirkby et al., 2016; Lehtipalo et al., 2018; Tröstl et al., 2016). Once growing past the critical sizes (e.g.,

50 nm to 100 nm), these newly formed particles can be activated as cloud condensation nuclei (CCN),

which in turn influence cloud formation and have climatic effects (Kerminen et al., 2005). Additionally,

NPF is a dominant source of atmospheric ultrafine particles in polluted urban environments (Yan et al.,

2021). These small particles (< 50 nm) can penetrate into the respiratory system, thus posing health risks

to human beings (Chen et al., 2016; Downward et al., 2018). Therefore, understanding NPF processes is



important both in terms of predicting climate change and understanding the health risks of aerosols (Kulmala et al., 2022).

Due to its chemically complex nature, the understanding of the key precursor vapours and controlling

mechanisms of urban NPF is still limited. Gaseous sulfuric acid and dimethylamine (DMA, $C_2H_7N$) are believed to play important roles in aerosol nucleation in urban environments (Xiao et al., 2021; Yao et al., 2018; Cai et al., 2021d; Almeida et al., 2013; Cai et al., 2022b). A recent study quantitatively demonstrated the decisive role of $H_2SO_4$ in initiating nucleation with the presence of stabilisers such as amines and $NH_3$ in Beijing (Yan et al., 2021). The subsequent growth of fresh particles is contributed

both by $H_2SO_4$ and oxidised organic vapours depending on the particle sizes. In urban Beijing, it was suggested that $H_2SO_4$ and its clusters contribute significantly to the growth of 1.5-3 nm particles (Deng et al., 2020b) while gas-phase oxygenated organic molecules (OOMs) promote the growth of 3-25 nm particles (Qiao et al., 2021).

Besides these widely studied species, oxidized iodine compounds were also found to introduce rapid

particle formation, mostly observed in mid-latitude coastal sites (Hoffmann et al., 2001; Mäkelä, 2002; O'dowd et al., 2002). Iodine nucleation was conventionally thought to be initiated by iodine oxides (Jimenez, 2003; Gomez Martin et al., 2020; O'dowd and Hoffmann, 2005; Hoffmann et al., 2001; O'dowd et al., 2002). However, field observations at the Mace Head observatory and dedicated experiments carried out in the CLOUD chamber at CERN revealed iodine oxoacids ($HIO_x$, i.e., $HIO_3$ and $HIO_2$ in this

study) as the key nucleating species in pristine regions (Zhang et al., 2022; He et al., 2021b). With state-of-the-art mass spectrometric methods, iodine oxoacids were recently identified in locations other than mid-latitude costal sites, such as in Arctic sites (Sipilä et al., 2016; Baccarini et al., 2020; Beck et al., 2021; He et al., 2021b), Antarctica sites (Jokinen et al., 2018; He et al., 2021b), boreal forest sites (Jokinen et al., 2022; He et al., 2021b), a remote marine site (He et al., 2021b) and importantly also in

polluted urban sites (He et al., 2021b). Chamber experiments have shown that $HIO_3$ (with $HIO_2$) nucleates faster than $H_2SO_4$ with 100 pptv $NH_3$ at the same temperature and equal acid concentrations, although iodine oxoacid nucleation rates are still lower than $H_2SO_4$-DMA nucleation (He et al., 2021b). It is worthwhile to note that the nucleation involving both iodine oxoacids and DMA remains unclear and iodine oxoacid nucleation may further be enhanced by strong bases (e.g., different amines) in urban

environments. After the formation of fresh particles, $HIO_3$ dominates the growth of iodine particles



between 1.8 and 3.2 nm at growth rates equal to those of $H_2SO_4$ (He et al., 2021b). It can be expected that, iodine oxoacids will contribute at least to sub-3 nm particle growth, and potentially also to particle nucleation, in polluted urban environments. Therefore, iodine oxoacids have the potential to enhance the survival probability (Kulmala et al., 2017) of fresh particles in the urban environment.


In order to quantitatively understand the contribution of iodine oxoacids in urban particle formation, we conducted a long-term measurement of iodine oxoacids and sulfuric acid ($H_2SO_4$) in urban Beijing from 2019 to 2021, and in suburban Nanjing from March 2019 to February 2020. Diurnal and seasonal trends of iodine and sulfur oxoacids are analysed and the potential sources of the unexpected iodine oxoacids are discussed. Moreover, we quantitatively discuss the contribution of $HIO_3$ to aerosol growth rate below

3 nm ($GR_{<3nm}$) and the potential enhancement in particle survival probability. Our study provides the first long-term observations of iodine oxoacids in polluted urban environments which could contribute to aerosol formation studies in inland cities.

## 2 Methods

### 2.1 Measurement sites and instruments

### 2.1.1 Sites

The measurement in urban Beijing was conducted from January 2019 to October 2021. The site locates on the fifth floor of the teaching building at the west campus of Beijing University of Chemical Technology (Aerosol and Haze Laboratory (AHL)/BUCT station, 39 ˚ 56´N, 116 ˚ 17´E). Located about

150 km away from the nearest coastline in the southeast, the station is surrounded by residential buildings and three main roads and a detailed description of this site can be found in a previous study (Liu et al., 2020). The observations in Nanjing were conducted at the Station for Observing Regional Process of Earth System (SORPES; 118˚57´E, 32˚07´N), a research and experiment platform inside Nanjing University, Xianlin Campus, 20 km northeast of downtown Nanjing and about 190 km away from the

nearest coastline in the east. Because of its unique geophysical location, the SORPES is considered to be a regional background station under the influence of anthropogenic plume from YRD (Yangtze River Delta) city cluster and multiscale transport coupled with Asian monsoon (Ding et al., 2016).



### 2.1.2 Acid concentrations

Gaseous iodine oxoacids (HIO$_3$ and HIO$_2$) and H$_2$SO$_4$ were detected by the nitrate-CIMS (Aerodyne

Research Inc. and Tofwerk AG) composed of a chemical ionization (CI) source and an atmospheric

pressure interface time-of-flight mass spectrometer (APi-TOF). Two long time-of-flight mass analysers

(LToF, resolution at around 10,000 Th Th$^{-1}$) were used at the AHL/BUCT station from January 2019 to

October 2021 and at SORPES station from March 2019 to December 2019, respectively, while a lower

resolution time-of-flight analyser (HToF, resolution at around 5,000 Th) was utilized at the SORPES

station from January 2020 to February 2020. As the comprehensive description of nitrate-CIMS has been

given in previous works (Junninen et al., 2010; Jokinen et al., 2012), they are only briefly discussed here.

Ambient air was drawn into a laminar flow reactor through a 0.75 in. diameter stainless steel tube with a

sample flow of about 7.2 L/min and surrounded by a purified airflow of 32 L/min serving as the sheath

flow at the AHL/BUCT station and 25 L/min at the SORPES station. The dominant reagent ions were

nitrate ions (NO$_3^-$ and HNO$_3 \bullet$ NO$_3^-$ and HNO$_3$HNO$_3 \bullet$ NO$_3^-$), which were generated in the sheath flow

by exposing gaseous nitric acid in the sheath flow to a photo ionizer X-ray (Model L9491, Hamamatsu,

Japan). The data of nitrate-CIMS were acquired at 1 Hz time resolution and analysed with the MATLAB

(MathWorks Inc.) toolbox ToFTools package (version 6.11) (Junninen et al., 2010).

### 2.1.3 Particle number size distribution

The particle number size distribution (PNSD) ranging from ~1 nm to 10 μm was measured using a home-

made diethylene glycol scanning mobility particle spectrometer (DEG-SMPS, 1-4.5 nm) (Jiang et al.,

2011) equipped with the miniature cylindrical differential mobility analyser (mini-cy DMA) (Cai et al.,

2017a) and a home-made particle size distribution system (PSD, 3 nm-10 μm) (Liu et al., 2016),

respectively at AHL/BUCT station, whereas two Scanning Mobility Particle Sizers (SMPSs) equipped

with a TSI longDMA (TSI Inc., model 3081) and a TSI nanoDMA (TSI Inc., model 3085), respectively

and Aerodynamic Particle Sizer (APS, TSI, APS-3321, USA, 500-1000 nm) were deployed to measure

PNSD at SORPES station. Additionally, ions of sizes range from 0.8 nm to 42 nm were measured using

a Neutral cluster and Air Ion Spectrometer (NAIS, Airel Ltd., Estonia) (Manninen et al., 2016) in Nanjing.



### 2.1.4 O₃ concentration and other meteorological factors

The ozone (O₃) concentration was measured using ozone analysers (49i, Thermo Fisher Scientific Inc. USA) at both sites. Additionally, ambient meteorological factors, including temperature (T), relative humidity (RH), and ultraviolet B radiation (UVB) were measured using an Automatic Weather Station (AWS310, Vaisala Inc.) in Beijing, whereas T, RH, and downward short-wave radiation (DSR) were recorded by sensors at the height of 44 m above the ground level at the SORPES station. The T and RH

were measured by a temperature and relative humidity probe (HMP155A, Campbell Inc., USA), and the DSR was recorded by a CNR4 net radiometer (OTT Hydromet Corp. Germany).

### 2.2 Data analysis

### 2.2.1 Characteristic of NPF events from PNSD

According to a widely used method (Kulmala et al., 2012), we classified all of the measurement days

into NPF and non-NPF events at both sites. All undefined days were regarded as non-NPF events in this study. Since there were periods when some key instruments failed to work, the NPF frequencies in each month were calculated as the ratio of the NPF event days to the days with valid data. The monthly statistics at both sites were summarized in Table S1. From the measured particle number size distribution, we calculated the condensation sink (CS) (Laakso et al., 2004; Kulmala et al., 2012), coagulation sink

(CoagS) (Kulmala et al., 2001), and growth rate (GR) for the NPF events.

CS, which characterises the loss rate of gaseous precursors and clusters onto the particles (Lehtinen et al., 2003) was calculated using the equation Eq. (1) (Kulmala et al., 2012):

$$CS = 4\pi D \sum_j \frac{1}{2} d_{p,j} \beta_m(Kn_j, \alpha) N_j \ , \tag{1}$$

where, $D$ is the H₂SO₄ vapour diffusion coefficient; $d_{p,j}$ is particle diameter; $\beta_m$ is transitional correction factor for mass flux as a function of $Kn_j$ (Knudsen number) and $\alpha$ (mass accommodation coefficient, assumed to be unity in this work) as shown in Eq. (2); $N_j$ is the number concentration of

$d_{p,j}$, and the particle diameter is corrected for growth factor according to T and RH (Laakso et al., 2004).



$$\beta_m = \frac{1+Kn_j}{1+0.377Kn_j+1.33Kn_j(1+Kn_j)/\alpha} \,, \tag{2}$$

To quantify if notable growth is to occur, especially at sizes below a few nanometers, it is crucial to
understand the loss process of fresh particles. Coagulation scavenging of freshly formed particles into
pre-existing particles before growing to significant sizes is essential for estimating the concentration of
newly nucleated particles at the size of 1.5-2 nm (Kulmala et al., 2001). Aerosol coagulation sink (CoagS)
represents this kind of coagulated scavenging characteristics. CoagS (the loss through coagulation among
particles) was determined from Eq. (3). Here, $K_{ij}$ is the coagulation coefficient (Kulmala et al., 2001).

$$CoagS = \sum_j K_{ij}N_j \,, \tag{3}$$

Besides, the GRs in NPF events were determined with both the appearance time method (including APT-
$x$ and APT-$y$) and the mode fitting method (MOD) to minimize the uncertainty from calculations
(Kulmala et al., 2012; Dada et al., 2020). Detailed approach is shown in supplementary materials. The
size-segregated GRs were calculated in two size ranges, i.e., sub-3 nm (GR$_{<3}$) and 3-7 nm (GR$_{3-7}$) based
on the appearance time method, and the 50% appearance time is fitted by smoothing the normalized
concentration timeseries for the particle of each size bin (Lehtipalo et al., 2014; He et al., 2021a). After
determining the 50% appearance time for each size bin, the GRs were fitted using the linear least square
method both with time as $x$ and $y$ to compare with each other and minimize the error. They are referred
to as APT-$x$ and APT-$y$, respectively in this study. The slope of particle size to their 50% appearance
time was regarded as GR using APT-$x$, which is the traditional way. However, as the particle diameter is
exactly measured by our instruments and the 50% appearance time is the independent variable
determined by calculations, we also tried to use the latter as the independent variable to fit the GR. In
this case, the GR was determined as the inverse of the fitted slope. The mode fitting (MOD) method fits
the particle number size distribution to find the mode diameters at any given time and tracks the evolution
of particle sizes. Up to now, there is still a debate about whether to adopt the appearance time method or
the mode fitting method for GR calculation, as neither is perfect for calculating GR for ambient
observations (Qiao et al., 2021; Deng et al., 2020b). For example, it is difficult to define the accurate
mode diameter, especially for sub-3 nm particles when the new particle formation just occurs. Therefore,



there could be some underestimation while using mode fitting method to calculate GR$_{<3}$ (Cai et al., 2022a). Determining the sub-3 nm particle growth can also be difficult for the 50% appearance time method for similar reasons. Additionally, appearance time method might be more sensitive to other processes as it does not track the growth of a particle or a population. (Lehtipalo et al., 2014; Cai et al., 2021c; He et al., 2021a). In this study, we report results using both methods to reduce the overall uncertainty of GR calculation and to provide a confidence range of GR. In both cases, the GR is determined from the rate of change in diameter shown as Eq. (4) (Kulmala et al., 2012).

$$GR = \frac{dd_p}{dt} ,$$  (4)

It is worthwhile to note that we corrected the GR obtained from the 50% appearance time method for the impact of coagulation sink, following Eq. (5) (Cai et al., 2021c).

$$GR_{corr,cond} = GR_{conv} - \left(CoagS + \frac{CoagSrc}{2N_p}\right) \times \left[\sqrt[3]{(d_p^3 + d_1^3)} - d_p\right] - GR_{coag} ,$$  (5)

where the $GR_{conv}$ is the GR calculated from conventional appearance time method in nm·s$^{-1}$; CoagSrc is the coagulation source defined as the production rate of the particle size bin because of coagulation, cm$^{-3}$s$^{-1}$, calculated using the Eq. (6); $N_p$ is the number concentration of particles with the size $d_p$; $GR_{coag}$ is the coagulation growth rate in nm·s$^{-1}$ from Eq. (7). More specific details can be found in Cai et al. (2021c).

$$CoagSrc = 0.5 \times \iint_{d_{p,1}^3 \leq d_i^3 + d_j^3}^{d_i^3 + d_j^3 \leq d_{p,u}^3} \beta_{i,j} n_i n_j \times d\log d_i \times d\log d_j ,$$  (6)

$$GR_{coag} = \sum_{d_p=d_{min}}^{d_p=d_p} \left\{\beta_{p,i} N_i \times \left[\sqrt[3]{(d_p^3 + d_i^3)} - d_p\right]\right\} ,$$  (7)

The counterbalance of CoagS and GR considerably affects the survival of small clusters. Survival probability (SP) is utilized to quantify the competition between growth and scavenging mentioned above (Veli-Matti Kerminen, 2002; Kerminen et al., 2005). We defined the SP$_{1.5-3}$ and SP$_{3-7}$ as the likelihood that the particles can grow from the smaller sizes to the larger sizes (i.e., from 1.5 to 3 nm and from 3 to



7 nm, respectively) before they are scavenged by coagulation. The SP can be calculated following Eq.

(8) (Lehtinen et al., 2007).

$$SP = \exp\left\{\frac{d_{p1}}{m-1}\frac{CoagS}{GR}\left[\left(\frac{d_{p2}}{d_{p1}}\right)^{1-m} - 1\right]\right\} , \qquad (8)$$

where $d_{p1}$ and $d_{p2}$ are the lower limit size and upper limit size, respectively; CoagS is the coagulation

sink at the lower limit size; GR is the averaged growth rate in the size range, from both corrected

appearance time method and mode fitting method; *m* was assumed to be 1.7 according to the measured

PNSDs (Lehtinen et al., 2007).

### 2.2.2 Contribution of HIO₃ to GR and SP

The particle growth rate due to HIO₃ concentration was observed to be linear in the CLOUD experiment,

shown as Eq. (9), which is fitted at 10 °C (He et al., 2021b):

$$GR(\text{HIO}_3)_{1.8-3.2} = 10^{log_{10}[\text{HIO}_3]-6.75} , \qquad (9)$$

where [HIO₃] is the iodic acid concentration in molecules cm$^{-3}$ and $GR(\text{HIO}_3)_{1.8-3.2}$ is the growth rate

of 1.8 to 3.2 nm particles in nm h$^{-1}$. However, the size range of sub-3 nm particles used in this study (1.5

to 3 nm) slightly differs from 1.8 to 3.2 nm and this formulation does not provide the growth rates of 3

to 7 nm particles. Additionally, as the temperature in summer seasons in both Beijing and Nanjing

(around 22 to 36 °C) is much higher than 10 °C, additional temperature correction is needed. In this study,

we adopt the equation provided by Nieminen et al. (2010) for these corrections:

$$GR'(\text{HIO}_3) = \frac{\Delta d_p}{\Delta t} = \frac{\Delta d_p \text{HIO}_3 \alpha_m m_v}{2\rho_v d_v} \cdot \sqrt{\frac{8kT}{\pi m_v}} \cdot \frac{1}{\left[\frac{2x_1+1}{x_1(x_1+1)} - \frac{2x_0+1}{x_0(x_0+1)} + 2ln\left(\frac{x_1(x_0+1)}{x_0(x_1+1)}\right)\right]} , \qquad (10)$$

where the subscript "*v*" refers to HIO₃. Additionally, $x_0$ and $x_1$ are the ratios of the diameter of HIO₃

molecule divided by the particle diameter at which the initial growth occurs (e.g., 1.5 nm or 1.8 nm) and

particle diameter at which the particles grow to (e.g., 3 nm or 3.2 nm), respectively. Two sets of growth

rates were calculated using this equation: 1) the first set utilized the measured ambient temperature at the

given growth period of NPF events with 1.5 to 3 nm or 3 to 7 nm as the growth ranges and 2) the second





set calculated the growth rates at 10 °C with 1.8 to 3.2 nm as the growth range (the same as at CLOUD).

The ratios of the growth rates calculated by 1) and 2) therefore give the correction factors that can be

applied to Eq. (9) to correct the temperature and size differences.

To quantify the growth rates of particles with mean diameter from around 1 nm to 7 nm in Nanjing, the

negative ion number size distribution collected by NAIS was utilized. However, it is extremely difficult

to use the NAIS to capture sub-3 nm particle growth rates as the limited atmospheric ions are mostly

captured by larger particles in polluted urban environments and thus leaving the sub-3 nm particle growth

undetectable (see supplementary materials for details). Therefore, in all NPF cases occurred at the

SORPES station, the contribution of gaseous iodic acid to sub-3 nm growth is only quantified by

comparing its contribution with that of sulfuric acid during the same event, since $H_2SO_4$ is believed to be

the dominant vapour for particle initial growth in sub-3 nm range (Deng et al., 2020b). $H_2SO_4$

contribution to GR is calculated as a first-order approximation as Eq. (11) (Stolzenburg et al., 2020),

where $d_p = \frac{d_{p_{initial}} + d_{p_{final}}}{2}$ in nm and $[H_2SO_4]$ is the gas phase $H_2SO_4$ concentration in molecule cm$^{-3}$.


$$GR(H_2SO_4) = \left(2.68 \times d_p^{-1.27} + 0.81\right) \times \left([H_2SO_4] \times 10^{-7}\right) , \tag{11}$$

We define $SP_{tot}$ as the particle survival probability calculated using the measured GRs (in Beijing) or the

expected growth rate considering growth contributions from both $H_2SO_4$ and $HIO_3$ (in Nanjing). In order

to quantify the SP enhancement by $HIO_3$, we further define $SP_1$ which represents the calculated survival

probability using GRs after deducting the growth contribution from $HIO_3$. Therefore, the enhancement

factor (EF) of *SP* can be represented as

$$EF = \frac{SP_{tot}}{SP_1} - 1 . \tag{12}$$


### 2.2.3 Iodic acid (HIO₃) precursor proxy

In order to investigate the source of gaseous $HIO_3$ at both sites, a daytime proxy formula is built to

describe the precursor level of measured $HIO_3$, which is as follows:

$$Proxy_{pre} = \frac{[HIO_3] \times CS}{UVB} \tag{13}$$



Eq. (13) is derived by assuming the $HIO_3$ concentration to be at a pseudo-steady state (the production

rate equals to the loss rate). Based on current knowledge about $HIO_3$ formation pathways, the proxy is

not intended to elucidate the composition of species serving as $HIO_3$ precursor or the related reactions.

Instead, it considers the photochemical reaction as the daytime formation pathway and condensation onto

pre-existing aerosol particles as the only sink for gaseous $HIO_3$.


## 3 Results and Discussion

### 3.1 Overview of the measurement

The measurement overviews in both Beijing and Nanjing are presented in Fig. 1, including the timeseries

of T, $O_3$, $HIO_x$, and $H_2SO_4$ concentrations, as well as the frequency of NPF events in each month. It

should be noted that each point on timeseries panels refers to daytime mean value. In this work, daytime

duration is defined between 08:00 and 16:00 in local time (UTC+8) considering the preferred time

window of NPF events in China (Kulmala et al., 2021), as shown in Fig. S2.

In Fig.1(a)/(d), it is obvious that the seasonal patterns of T and $O_3$ are similar during measurement periods

in both sites, i.e., both peak in the summer. The $O_3$ levels are roughly the same at both sites and the

maximum values of daily mean T are both over 35℃, though the lowest T (about 1℃) in Nanjing is

significantly higher than that in Beijing (about -12℃).

$H_2SO_4$ concentration is slightly lower in cold seasons (Fig.1(b)/(e)). $H_2SO_4$ concentration exceeds $10^7$

$cm^{-3}$ only on a few days in Beijing, whereas it is a common phenomenon in Nanjing daytime. Besides

$H_2SO_4$, we report the first long-term measurement of $HIO_x$ in urban environments continued from earlier

sparse measurements (He et al., 2021b). The calibrated $HIO_x$ concentration is above the detection limit

during almost the entire measurement periods, indicating a clear presence of $HIO_x$ in inland cities. The

$HIO_3$ concentration was between $10^5$ and $10^6$ $cm^{-3}$ for most of the time except for winter months; it

sporadically approaches or is higher than $10^6$ $cm^{-3}$ in warm months. On the other hand, iodous acid, $HIO_2$

is less abundant than $HIO_3$ at both sites with a general concentration at around $10^4$ $cm^{-3}$ and a maximum

concentration approaching $10^5$ $cm^{-3}$ in the summer. The results indicate that the $H_2SO_4$ concentrations

are generally higher than that of iodine oxoacids at both sites. As for the two iodine oxoacids ($HIO_3$ and

$HIO_2$), daytime mean concentration of $HIO_3$ is more than one order of magnitude higher than $HIO_2$. The



one order of magnitude lower $HIO_3$ concentration compared with $H_2SO_4$ in summer at both sites is consistent with that in the Finnish subarctic boreal forest (Jokinen et al., 2022).

The frequencies of NPF events varied significantly, from none to more than three quarters of the days in each month during the measurement period. The occurrence of NPF events in China is favoured by various meteorological factors (Qi et al., 2015; Zhou et al., 2021; Chu et al., 2019). However, the influences can be quite uncertain and complex because of different season and the location of measurement site. Take temperature for an example, on one hand, warm temperatures enhance the

abundance of biogenic and anthropogenic volatile organic compound emissions as well as their oxidation processes (Paasonen et al., 2013; Paasonen et al., 2018; Nie et al., 2022; Ehn et al., 2014). On the other hand, the warm temperature also reduces the stability of embryonic clusters thus reducing nucleation and subsequent growth rates (Kürten et al., 2016). Besides meteorological conditions, vapour condensation sink (CS) and particle coagulation sink (CoagS) have negative effects on the NPF frequency

(Kalkavouras et al., 2017; Bousiotis et al., 2021; Wehner et al., 2007). Decreased CS and/or CoagS will lead to faster nucleation and subsequent growth (as scavenging of nucleating and condensable vapours is less effective) and higher survival probability through the growth processes during NPF events (as the scavenging of clusters and small particles is less effective). As expected, the concentration of gaseous $H_2SO_4$ is notably correlated with NPF frequency, as $H_2SO_4$ is the most important compound to form

initial clusters and one of the main contributors to the growth of newly formed particles (Nieminen et al., 2010; Kirkby et al., 2011). During this measurement, the ratio of $GR_{1.5-3}$ contributed from $H_2SO_4$ to measured GR calculated from MOD is about 72.4% (shown in Table S4).

As depicted in Figure 1(c)/(f), the frequencies of NPF for each month at two sites are quite different,

since environments are chemically complex and diverse with many aforementioned factors influencing NPF. Generally, NPF events are more likely to occur in the spring and winter, with the lowest frequency in summer at BUCT station in Beijing, consistent with other reports (Wu et al., 2007; Deng et al., 2020b). Different from Beijing, there are less NPF events in the winter than in the summer at SORPES station, which is in line with a long-term measurement conducted at the same site (Qi et al., 2015). It could be

attributed to the lowest $H_2SO_4$ concentration in cold season, which was found to be the main driver for NPF events in polluted megacities in China (Yao et al., 2018). Another explanation may be that the high CS in the winter daytime (Qi et al., 2015) suppresses the NPF events. It should be noted that the particle



formation mechanism in Nanjing is yet to be revealed and NPF intensity could be reduced if DMA is limited in Nanjing.

**3.2 Characteristic of acid concentrations**

**3.2.1 Seasonal variation**

To better understand the roles of the studied acids in new particle formation and growth, we further present the seasonal variation of $H_2SO_4$, $HIO_3$ and $HIO_2$ concentrations in Fig. S3. It depicts the monthly statistics of $H_2SO_4$ and $HIO_x$ at the two sites with different shadings indicating seasons. Both $H_2SO_4$ and

$HIO_x$ concentrations in Nanjing are always higher than those in Beijing except for $H_2SO_4$ concentrations in the winter. We speculate that the generally higher acid concentrations in Nanjing are caused by stronger solar radiation at latitude of 32°07′N in Nanjing, compared with 39°56′N in Beijing. On the other hand, the reason of higher wintertime $H_2SO_4$ concentrations in Beijing is likely due to the higher $SO_2$ and more frequent sunny weather during the winter in Beijing (Wang et al., 2018) and further

discussion can be seen in supplementary materials. At both sites, the seasonal pattern of $H_2SO_4$ is not very strong (Deng et al., 2020b; Petäjä et al., 2009). $H_2SO_4$ concentrations are higher in the spring and autumn, lower in the summer, and lowest in the winter.

The $HIO_3$ concentrations measured at the two sites are significantly lower than that at pristine coastal site (e.g., in Mace Head (Sipilä et al., 2016)). Measurements at Mace Head indicate that $HIO_3$

concentrations are frequently above $10^7$ cm$^{-3}$ with some days exceeding $10^8$ cm$^{-3}$ in September. The concentrations of atmospheric iodine at coastal sites are normally higher due to active biogenic emissions of iodine-containing precursors from marine algae (O'dowd et al., 2002). On the other hand, $HIO_3$ concentrations in Beijing and Nanjing are comparable to that in Helsinki, Finland. Measurements at SMEAR III station, an urban site located in University of Helsinki show $HIO_3$ concentrations at around

$10^6$ cm$^{-3}$ when the wind is coming from land for most times in August and $HIO_3$ concentrations exceed $10^7$ cm$^{-3}$ when air masses have marine origin (Thakur et al., 2022; He et al., 2021b). Another long-term observation conducted at SMEAR I station (Jokinen et al., 2022), a subarctic boreal forest site, shows $HIO_3$ concentrations often at around $10^5$ cm$^{-3}$ from April to November 2019 (summer and autumn) with occasional peaks exceeding $10^6$ cm$^{-3}$ in late August. $HIO_x$ concentrations at AHL/BUCT station depict a

distinctly unimodal pattern in a year cycle with highest values in July, increasing from January and decreasing to December. However, seasonal variations of $HIO_x$ are slightly different at SORPES, as there



are similar levels of HIO$_x$ throughout the summer in 2019, reaching a seemingly steady daily maximum. HIO$_3$ concentration measured from August to September in 2018 over the central Arctic Ocean increases significantly from summer towards autumn (Baccarini et al., 2020), which is different from the results in

both Beijing and Nanjing, due to significantly different environments and iodine sources.

### 3.2.2 Diurnal pattern

Though the HIO$_3$ concentrations are different in four seasons, the diurnal patterns are similar throughout the year (Fig. 2). Daily trends of both median H$_2$SO$_4$ and HIO$_x$ concentration are strongly connected with

diurnal cycle. The concentration of HIO$_x$ increases at the same time as H$_2$SO$_4$, i.e., both HIO$_x$ and H$_2$SO$_4$ rise in the early morning and peak from noon to afternoon. The clear diurnal pattern of H$_2$SO$_4$ has been attributed to photochemical activities (Lu et al., 2019; Yang et al., 2021; Petäjä et al., 2009).  Hydroxyl radical (OH) is the most important oxidant for sulfur dioxide (SO$_2$) to form daytime H$_2$SO$_4$ (Guo et al., 2021; Yang et al., 2021). Therefore, the diurnal pattern of H$_2$SO$_4$ would be affected by its precursors (e.g.,

SO$_2$ and OH in daytime). Higher HIO$_x$ concentrations in the daytime and the absence of their nocturnal maxima suggest that the main source of HIO$_x$ is also photochemical oxidation of iodine precursor vapours. The distinct diurnal variation in HIO$_x$ concentration with around one order of magnitude implies fast in-situ chemistry. Although the diurnal patterns of H$_2$SO$_4$ and HIO$_3$ are alike, the occurrence of HIO$_3$ daytime maximum is on average later than that of H$_2$SO$_4$ at both sites (Fig. 2). This phenomenon is

pronounced during summer daytime when daily maximum of H$_2$SO$_4$ appears around 2 hours earlier than that of HIO$_3$. It implies that albeit these two acids form during daytime through photochemical pathways, the limiting factors for their productions can be different. At the SORPES station, for instance, the diurnal cycle of H$_2$SO$_4$ follow that of radiation in spring, summer and winter. In summer, however, owing to effectiveness of long-term emission reduction, SO$_2$ concentrations can be low enough to limit the

production of H$_2$SO$_4$ (Ding et al., 2019), so the daytime peaks of H$_2$SO$_4$ tend to occur when SO$_2$ reached its daily maximum (Yang et al., 2021). On the other hand, little is known about the diurnal patterns of HIO$_x$ in urban environments. It was demonstrated in chamber experiments that HIO$_x$ can be formed by oxidation of oxidised iodine species with ozone in the absence of HO$_x$ (He et al., 2021b) and I$_2$O$_2$ + O$_3$ reaction was recently found to be the critical step for the HIO$_3$ formation (Finkenzeller et al., 2023). The

authors found that ambient level of O$_3$ was not the limiting factor for HIO$_3$ formation. However, the



maximum of $HIO_3$ was found to mimic that of $O_3$, indicating that $O_3$ may influence terrestrial $HIO_3$ formation. One such possibility is that the source of iodine is controlled by $O_3$ deposition, e.g., a similar process to that occurring over marine surfaces (Carpenter et al., 2013). This mechanism mainly explains the inorganic iodine emissions from marine environments, but a similar process may also occur in

polluted urban environments although the iodine source in urban environments is yet to be discovered. Another possibility is that air temperature may strongly perturb the formation of $HIO_3$ and the release of iodine precursors, which will be discussed in detail in the next section.

Moreover, CS in Nanjing shows an opposite profile to T and $O_3$, whereas it keeps almost the same trend

as that of Nanjing but fluctuates a little during the day in Beijing spring and winter, which show median values only in 2019. In summary, the diurnal patterns shown in Fig. 2 suggest that stronger solar radiation coupled with higher mixing ratio of $O_3$ and higher T are likely the factors favouring the formation of acids, the low CS at noon is preferred for the survival of acid vapour. Additionally, the diurnal variation of $HIO_x$ at BUCT shows stronger seasonality with highest values at around noon in the summer. The

maximum concentrations at spring and autumn are similar, while the maximum concentrations in the winter are roughly one order of magnitude lower. At SORPES, $HIO_x$ reaches similar levels in the spring, summer and autumn but its concentration is lower in the winter. Consistently, the diurnal maximum $HIO_3$ concentration in summer approaches $10^6$ cm$^{-3}$ at both sites.

**3.2.3 Iodine sources**

In order to investigate the source of $HIO_x$ in urban environments, we further conduct cluster analysis of the air mass backward trajectories of the AHL/BUCT station. The $HIO_3$ precursor proxy calculated from Eq. (13) based on the measurement results at BUCT/AHL station is classified into four levels as shown in Fig. S4. High precursor levels are mainly associated with air masses originating from the south and

southeast, whereas lower iodine precursor concentrations are associated with northern air masses. It implies that marine iodine sources could be important for the AHL/BUCT station due to long range transport. Additionally, higher concentration of iodine precursor when the air mass travels from southern land indicates that the continental outflows may also play a significant role in transporting $HIO_3$ precursors. Therefore, both marine (O'dowd and Hoffmann, 2005; Carpenter et al., 2021) and terrestrial



precursors (Li et al., 2014; Wang et al., 2017) may contribute to the HIO$_3$ formation at the AHL/BUCT

site.

Both seasonal variation and diurnal pattern shows the lowest concentration of HIO$_x$ in winter when the

impact from residential coal burning and fossil fuel combustion power plant in Beijing is the largest. It

implies that HIO$_x$ concentration is not promoted by pollution in cold season. A previous 2-year

measurements conducted in Beijing show that high loadings of particulate organic iodine compounds

(OICs) occurred in the heating season, and HOI was thought to be the key oxidant to form the OICs (Shi

et al., 2021). The different seasonal distribution between gaseous HIO$_x$ in this study and particle-phase

OICs indicates potentially different iodine sources of gaseous and particulate phases, which warrants

further investigation. Figure 3a presents HIO$_3$ in different PM$_{2.5}$ ranges and shows lower HIO$_3$

concentrations when PM$_{2.5}$ increases. PM$_{2.5}$ measurements in the Beijing–Tianjin–Hebei region (2013 to

2020) show obvious seasonal characteristics with lowest concentrations in summer and highest

concentrations in winter (Yang et al., 2022). This phenomenon should be attributed to the inherently

independent seasonality of these two constituents instead of any correlation. If the PM$_{2.5}$ is a source of

gaseous iodine species, the HIO$_3$ concentrations should be higher in winter months, which is not the case.

Additionally, summertime HIO$_3$ concentrations in different PM$_{2.5}$ concentration bins have no difference

(Fig. 3b), which further indicates that the HIO$_3$ is not correlated with the particulate matter pollution in

Beijing.

There is no definite evidence to justify whether marine or land sources could better explain our

observation on HIO$_3$ concentration in Beijing. In the marine environments, the rapid reaction of sea-

surface iodide with O$_3$ is believed to be the largest global source of iodine species in the forms of

molecular iodine, I$_2$ and hypoiodous acid, HOI (Carpenter et al., 2021; Carpenter et al., 2013), which in

turn contributes to the formation of HIO$_x$ (Finkenzeller et al., 2023; He et al., 2021b). However, the

photolysis lifetimes of HOI (~140 s) and I$_2$ (~10 s), or biogenic volatile iodocarbons (e.g., CH$_2$I$_2$ (~ 5

min)) are too short to contribute to the formation of HIO$_x$ in Beijing and Nanjing considering the long-

range transportation (Saiz-Lopez et al., 2012). Another iodine-containing species, methyl iodide (CH$_3$I),

has a longer lifetime of about 5 days which may potentially go through the long-range transportation and

eventually reach inland cities. CH$_3$I is dominantly formed from photochemical processes in the marine



surface (Moore and Zafiriou, 1994) and additionally also from dust stimulated abiotic emission (Williams et al., 2007). $CH_3I$ concentration was shown to be correlated with surface seawater temperature (SST) in marine boundary layer air at midlatitude (Yokouchi et al., 2008). Others reported the opposite results in the Yellow Sea and the East China Sea during summer (Li et al., 2021) and the reason may be that higher surface water temperature also accelerates the chemical loss of $CH_3I$ from the seawater and atmospheric

$CH_3I$ is readily photolyzed. Long-term variations of atmospheric $CH_3I$ at several sites show that SST near each site cannot fully explain the variation of observed $CH_3I$ concentrations (Yokouchi et al., 2012). Other factors such as acidification, i.e., pH conditions, mineral dust deposition and dissolved organic carbon (DOC) concentration (Li et al., 2021), as well as ferric ion ($Fe^{3+}$) concentration (Chen et al., 2020) in seawater could also contribute to the emission rate of $CH_3I$.


Apart from marine sources, terrestrial sources of $CH_3I$ including most rice paddies (Redeker et al., 2000), terrestrial biomes (Sive et al., 2007), and minor wetlands (Dimmer et al., 2001), biomass burning (Andreae et al., 1996) were also proposed. High concentration of $CH_3I$ at two inland sites in Japan indicates the greater importance of terrestrial sources in the summer compared to oceanic sources

(Yokouchi et al., 2008). As $CH_3I$ emission from rice paddies is positively correlated with temperature (Redeker and Cicerone, 2004; Redeker et al., 2000), $CH_3I$ emission is likely to be stronger in the summertime. This is consistent with higher concentrations of $HIO_x$ in Beijing and Nanjing as shown in Fig. 1, 2 and 4. Moreover, experiments show that $CH_3I$ emission under dark incubation was much lower than that under light incubation and $CH_3I$ production under visible light conditions is lower than that

under natural light (Li et al., 2021). Those results indicate that ultraviolet light promotes the production of $CH_3I$ (Chen et al., 2020; Li et al., 2021) as well as its photochemical oxidation. This is consistent with our observation that $HIO_x$ is only formed in the daytime (Fig. 4).

The relative importance of terrestrial and marine iodine sources may vary widely with local

meteorological factors and the transportation of air masses. Future efforts are needed to verify the composition and distribution of $HIO_x$ precursors in polluted environments.



### 3.2.4 Formation of HIO₃ in urban environments

Quantum chemical methods and laboratory experiments have both been carried out to investigate the
formation mechanisms of $HIO_3$. Previous studies using quantum chemical calculations have proposed
two possible formation pathways of $HIO_3$. Iodine monoxide (IO) was proposed to react with $HO_2$ radical
to yield $HIO_3$ (Drougas and Kosmas, 2005). The reaction between iodine dioxide (OIO) and OH radical
was also suggested to produce $HIO_3$ (Plane et al., 2006). Laminar flow reactor experiments have also
been carried out to investigate the formation mechanisms of $HIO_3$ from $I_2$ (He, 2017). With the
illumination of green-fluorescent lamp, $I_2$ was efficiently photolyzed while $O_3$ photolysis was restricted,
and thus there was no known source of $HO_x$. Surprisingly, a significant amount of $HIO_3$ was formed
essentially under $HO_x$ free conditions. To rule out a potential unknown OH source in the flow reactor,
different OH scavengers (methane, sulfur dioxide, cyclohexene and acetic acid) were injected but the
production of $HIO_3$ remained. Iodine atoms or iodine oxides were proposed to be the reactants with ozone
and water to produce $HIO_3$. Following studies from the CLOUD experiments suggest that iodooxy
hypoiodite (IOIO) could be efficiently converted into $HIO_3$ via reactions R (1) and R (2) (Finkenzeller
et al., 2023) which successfully explain earlier laboratory and field observations (Sipilä et al., 2016; He
et al., 2021b). These results align with our observations at the AHL/BUCT station. UVB is an indication
of light intensity and influences the production of I atoms from the photolysis of $I_2$ and $CH_3I$. The
produced iodine atoms react with $O_3$ and further drive gaseous iodine chemistry (Saiz-Lopez et al., 2012).
Fig. 4 depicts the correlation of UVB, $O_3$ and temperature. Although the correlation between $HIO_3$
concentration and air temperature is not very strong, high $HIO_3$ concentrations appear when both the
UVB and $O_3$ mixing ratios are high. This is consistent with the fact that both solar radiation and $O_3$ are
required to initiate the iodine emission and iodine photochemistry.


$$IOIO + O_3 \rightarrow IOIO_4 \hspace{4cm} R\ (1)$$
$$IOIO_4 + H_2O \rightarrow HIO_3 + HOI + O_2 \hspace{2.5cm} R\ (2)$$

### 3.3 Iodic acid enhances the particle survival probability

Statistical results show that NPF events occur frequently at both sites throughout the measurement
periods. Whether freshly nucleated particles can contribute to cloud condensation nuclei and hence





impose influence on climate and human health depends largely on how fast they grow into larger particles and survive from coagulation scavenging by pre-existing aerosols, which is more efficient for smaller particles. The GR of newly formed particles is therefore central for sub-10 nm particle lifetimes in

ambient environment. Earlier studies have linked some dimensionless parameters L or $L_\Gamma$ (Mcmurry et al., 2005; Kuang et al., 2010; Cai et al., 2017b) to justify the occurrence of NPF and described the competition between aerosol surface area and condensable vapours during the growth period. Recently, a dimensionless survival parameter P (Kulmala et al., 2017) was proposed as the ratio of CS' (CS/$10^{-4}$ s$^{-1}$) to GR' (GR/1nm hour$^{-1}$). From this ratio, P shows the competition between the possibility of being

cleared and growing to survive. When P is below 50 in clean environment or 100 in polluted urban cities, the SP of the sub-3 nm particles is agreed with the atmospheric observations. As shown in Fig. S6, the median P is about 50 for the sub-3 nm particles, whereas the median P value is about 20 for 3-7 nm particles in the NPF events at the AHL/BUCT station. That means the growing particles in those days preferred to survive and thus showed us clear new particle formation and further growth.


As shown in Eq. (8), the impact of GR on survival of new particles is not linear and a small enhancement on GR could result in much larger enhancement in particle survival probability. To further illustrate the non-linear response of particle SP, we plot the logarithmic value of SP ($\log_{10} SP$) as a function of CoagS and GR from 1.5 to 3 nm (Fig. 5 panel a) and from 3 to 7 nm (Fig. 5 panel b), respectively. The value of

SP is extremely sensitive both to CoagS and GR. Under the typical CoagS (around 0.0025 s$^{-1}$) at both sites, the SP could be enhanced by more than two orders of magnitude when GR is varied from 1 to 10 nm h$^{-1}$. Increased GR caused by additional condensing vapours enables faster growth, which in turn facilitates the survival of sub-10 nm particles from coagulation scavenging. This effect is especially important for sub-3 nm particles as they are the most susceptible and are easily lost to large pre-existing

particles. A limited variation of GR was suggested to cause considerable variation of SP (Cai et al., 2021a).

We present case studies of several consecutive NPF events at both sites in Fig. 6. At AHL/BUCT station, NPF events occurred from May 25 to 29, 2021 due to favourable meteorological conditions, except for

one undefined day (May 27, 2021) when no obvious growth was observed. On this undefined day, the UVB in the daytime was low and the intensity was fluctuating, due to cloudiness. Both of these



conditions, as well as the higher CS, suppressed the NPF (Kerminen et al., 2018; Deng et al., 2020a; Cai et al., 2021b). On the other hand, both T and $O_3$, as well as UVB, increased from around 6:00 in the morning on the NPF event days, with decreasing RH. The averaged concentrations of $H_2SO_4$ and $HIO_3$

in the particle growth periods from 1.5 nm to 7 nm on event days and from 8:00 to 10:00 on the other days were summarized in Table 1. The ratio of $HIO_3$ concentration to $H_2SO_4$ was about 5% in the first three days and more than 10% in the next two days, likely due to higher $O_3$ concentrations which contributes to the emission of iodine precursors.

Table S2~S4 and S5~S7 summarise the $HIO_3$ contribution to GR and SP in the particle size range of 1.5-3 nm and 3-7 nm, respectively. To account for the uncertainties in the GR and SP calculations, the measured GR are calculated using three methods, namely the APT-$x$, APT-$y$ and the mode-fitting (MOD) methods (see Methods part, Fig. S7). We present the results from MOD methods in the main text to keep consistency with earlier studies (Deng et al., 2020b; Qiao et al., 2021). Only events with clear growth

were reported in this study to reduce systematic uncertainties resulting from the GR calculation. We also provide the results from the APT-$x$ and APT-$y$ methods in the supplementary materials for completeness. The results in Table S2-S4 show that the contributions of $HIO_3$ to $GR_{<3}$ on May 25 and 26 were lower than 5%, whereas the contribution was more than 10% on May 29. The $SP_{1.5-3}$ enhancement from $HIO_3$ is much stronger on May 29, even reaching 40.5%. Although the contribution of $HIO_3$ to $GR_{<3}$ on Jun

21, 2021 is almost identical to that on May 29, 2021, SP enhancements on Jun 21, 2021 is 2.5 times larger. This is a result from the 5 times CoagS on Jun 21, 2021. This suggests that in polluted environments with higher CoagS, such as Beijing, a fixed GR enhancement can lead to larger SP enhancements. Results in both Fig. 7 and Fig. S5 show that the median contribution of $HIO_3$ to the GR of particles in the 1.5-3 nm size range is 7.4% using the MOD method, whereas the contribution is only

around 3% and 2% using the APT-$x$ and APT-$y$ methods, respectively. This is resulted from the difference in the measured GR calculated using either the APT or the MOD methods. This further translates into 10.8%, 4.1%, 40.5% SP EF using MOD, APT-$x$ and APT-$y$ methods, respectively. Despite the uncertainty in the measurement GR calculation, the EF of $HIO_3$ to sub-3 nm particle SP is clear: in 33.3%, 25.0% and 55.6% of the events $HIO_3$ enhances particle SP by more than 30%, using MOD, APT-

$x$ and APT-$y$ methods, respectively.



Additionally, we show three consecutive NPF events observed from 16 to 18 August 2019 at the SORPES station, Nanjing. These events feature high acid concentrations, $O_3$ mixing ratios and strong light intensities as well as low RH and CS. No nucleation mode particles burst was observed on August 20 probably owing to decreased $H_2SO_4$ concentration. Compared to the cases in Beijing, the concentrations

of $O_3$ and acids are twice as higher at the SORPES station due to geographical and seasonal differences (Table 1).

From June to December 2019, there are 23 NPF events recognized at the SORPES station. Due to the detection limitation of the instruments at SORPES, the sub-3 nm particles were not clearly measured, which in turn poses challenges to get the measured GR through the 50% appearance time method nor the

mode fitting method. The statistics are depicted in a different way from that of the AHL/BUCT (see supplementary materials for further details). Briefly, the contribution of $HIO_3$ and $H_2SO_4$ to sub-3 nm particle GR in NPF events are quantified based on Eq. (9) ,Eq.(10) and Eq.(11). This is based on the observed consistency between the gaseous $H_2SO_4$ concentration and its significant contribution to the sub-3 nm particle growth rate in Beijing (Deng et al., 2020b). The SP enhancement is further calculated

based on Eq. (12). Consistent with the observation at the AHL/BUCT station, the concentration of gaseous $HIO_3$ in Nanjing is lower than $H_2SO_4$ throughout the measurement period, accounting for 10~20% of concentration (Fig. 1e). However, the average $HIO_3$:$H_2SO_4$ ratio (16.8%) is higher than that in Beijing (10.7%).

The calculated GR contribution of IA and SA to sub-3 nm particles ($Ratio_{1.5-3}$) are listed in Table S8,

respectively, where the statistical ratios of acid contributions for each NPF event are listed as well. The computations of Eq. (9) and Eq. (10) are subject to acid concentrations and hence the average acid concentrations for individual NPF events dominate the statistics of GR ratio. At the SORPES station from June to November 2019, $HIO_3$ contributes 6.1% (median) and 6.7% (mean) to sub-3 nm particles growth compared to $H_2SO_4$.


As listed in Table S9, estimated particle survival probability of particles growing from 1.5 to 3 nm considering $H_2SO_4$ as the governing contributor ($SP_{1.5-3}(SA)$) is significantly enhanced when counting the contribution from $HIO_3$ ($SP_{1.5-3}(SA+IA)$). The enhancement of SP with IA being an additional GR contributor varies from 3% to more than 100% in favourable cases, with a median enhancement of 54.3%.

For sub-3 nm particles, the survival probability is twice as higher (enhancement factors exceeding 100%)



considering $HIO_3$ as additional GR contributor on July 3 and October 26. As depicted in Fig. S8(c), SP enhancements in percentage are generally one order of magnitude higher than the GR contribution in percentage and $HIO_3$ can result in as high as 2-fold enhancement on SP in sub-3 nm particle growth.

The statistical results at both sites suggest that for polluted environments with higher CoagS, GR enhancement is especially important for the survival of small particles. In Beijing, $HIO_3$ contributes 7.4% (median) to sub-3 nm particle growth and 2.6% (median) to 3-7 nm particle growth in all NPF events from May to September. Despite the limited GR contribution, it could lead to 40.5% enhancement of $SP_{<3}$, whereas there is only a negligible increasement of 3.2% (median) for $SP_{3-7}$, estimated using the
MOD method. In exceptional cases, we found that $HIO_3$ could enhance particle SP by more than two-fold in 22.2% of cases. The median value of GR contribution of $HIO_3$ accounts for 6.1% of $H_2SO_4$ and the median enhancement of 1.5-3 nm particles survival probability reaches 47.6% when consider $HIO_3$ as an additional GR contributor, at the SORPES station in summer 2019. In favourable cases, the gaseous $HIO_3$ can contribute to more than 14.0% of particle growth, leading to the survival probability of fresh
particles enhanced by a factor of two. The role of the other iodine oxoacid, $HIO_2$, in particle growth remains unclear due to the absence of contribution equation like Eq. (10) for $HIO_3$. The estimated contribution of $HIO_2$ derived from the same equation should be significantly smaller compared with that of $HIO_3$ even if the arrival rate of $HIO_2$ also reaches the kinetic limit, as measured $HIO_2$ concentrations at both sites are much lower than $HIO_3$.

In summary, our findings show that $HIO_3$ is an important contributor to sub-3 nm particle survival in these two Chinese cities and similar environments elsewhere in warm seasons. However, for particles in 3-7 nm, the contribution of $HIO_3$ to particle GR and SP is negligible.

## 4 Conclusion

In this study, we show three years' measurements of iodine oxoacids ($HIO_x$) in Beijing and one-year
observation in Nanjing. Unlike $H_2SO_4$, $HIO_3$ has a more prominent seasonal variation at both sites with highest concentrations in summer and lowest concentrations in winter. The diurnal pattern of $HIO_x$ indicates that $HIO_3$ formation is influenced by photochemical activities and $O_3$ concentrations which may together influence the emission of iodine species and the further oxidation chemistry. In Beijing, back



trajectory analysis suggests that marine iodine sources are important for the $HIO_x$ production and less $HIO_x$ is observed if the air masses originate from the North. The lowest concentrations of $HIO_x$ in winter and its weak correlation with $PM_{2.5}$ implies that anthropogenic activities are likely not the important sources of $HIO_x$.

We find that the median contribution of iodic acid, $HIO_3$, to $GR_{sub-3}$ is less than 10% in Beijing and in Nanjing from May to September. However, $HIO_3$ can significantly enhance particle survival probability, occasionally by two-fold, for 1.5-3 nm particles at both sites. This means that although $H_2SO_4$ is considered to be the main driver of sub-3 nm growth in polluted urban areas, additional sources, such as $HIO_3$, needs to be considered. As the growth rate of $HIO_3$ is measured to be identical to that of $H_2SO_4$ on a per-molecule basis (He et al., 2021b), we propose that $HIO_3$ and $H_2SO_4$ can be summed up when estimating the sub-3 nm particle growth rates. Beside the enhancement on particle growth, recent theoretical studies have indicated that dimethyl amine could potentially accelerate pure $HIO_3$ nucleation (Ning et al., 2022). However, experimental confirmation is needed to confirm such prediction and our long-term observation can therefore provide basis for guiding experimental works to use ambient relevant level acid concentrations.




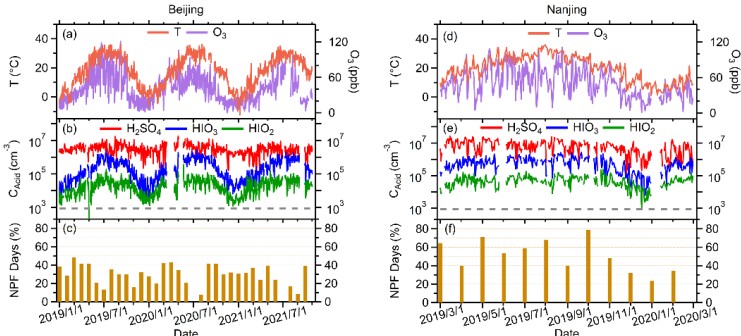

**Figure 1. Timeseries of parameters from Jan 1, 2019 to Oct 31, 2021 in Beijing (a-c) and from March 1, 2019 to Feb 29, 2020 in Nanjing (d-f). (a/d). Temperature and ozone; (b/e). Sulfuric acid and iodine oxoacid concentrations. The grey dashed line represents the detection limit of instruments (875 cm⁻³); (c/f). The frequencies of new particle formation events in each month. Time resolution for all the presented data is 1 day and the environmental parameters and vapour concentrations are averaged daytime (8 am to 4 pm) mean.**

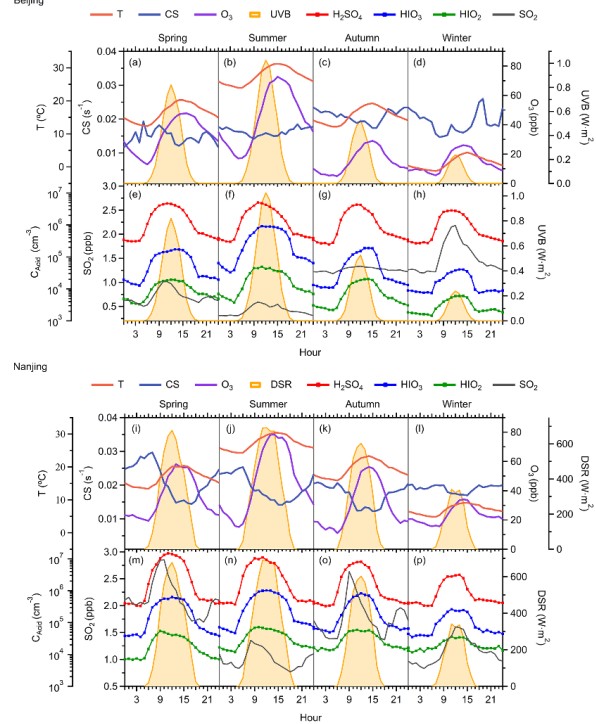

**Figure 2. Diurnal variation of median value of O₃, T, and CS in the first row (Beijing, a-d) and third row (Nanjing, i-j) and the diurnal variation of median SO₂, H₂SO₄ and HIOₓ concentrations in the second row (Beijing, e-h) and fourth row (Nanjing, m-p) in four seasons. The first to last columns are profiles in spring, summer, autumn, and winter, respectively. The diurnal patterns of UVB were plotted in every panel to compare with other factors better.**



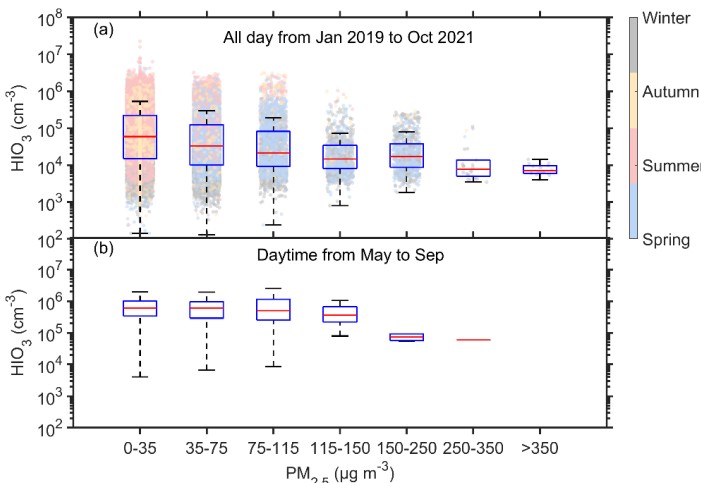

**Figure 3. HIO$_3$ concentration in different PM$_{2.5}$ level bins. (a) data from the whole campaign coloured by the seasons in Beijing and (b) data in the daytime from May to September (warm seasons) in Beijing.**

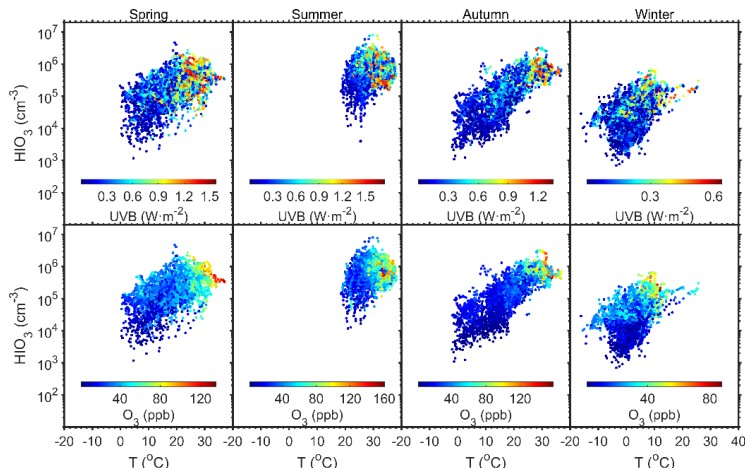

**Figure 4. Influences of T, UVB, and O$_3$ on HIO$_3$ concentration in the daytime (8:00-16:00) in Beijing. The analysis is separated into four seasons.**



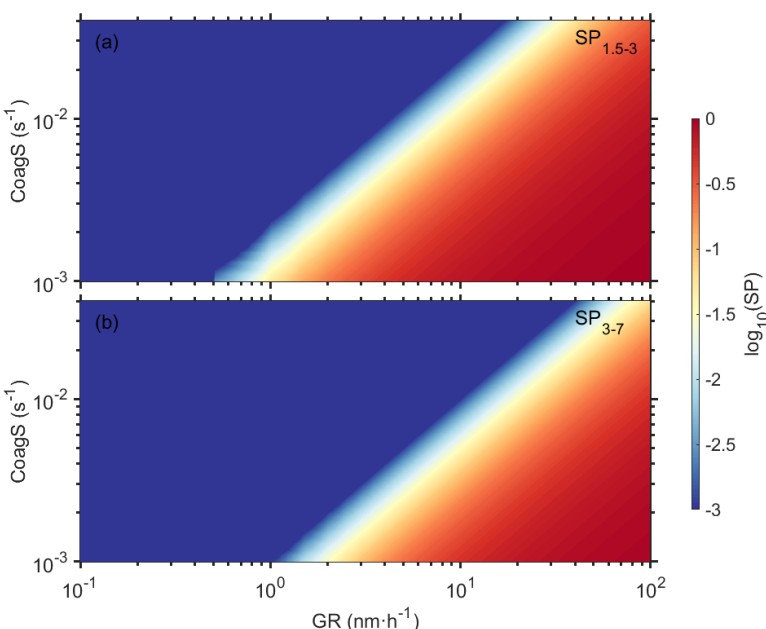

**Figure 5. The effect of coagulation sink and growth rate on particle survival probability for 1.5-3 nm (a) and 3-7 nm (b) particles, respectively.**

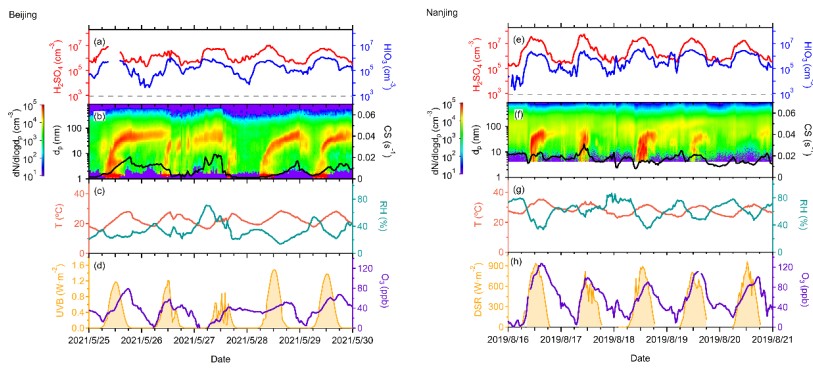

**Figure 6. Cases of consecutive NPF events in Beijing (a-d) and Nanjing (e-h) sites. Acid concentrations are shown in the first row, particle number size distribution, and CS are shown in the second row, meteorological factors, such as T, RH, UVB, and O$_3$ concentrations are also presented in the third and fourth rows. The measurement of acids in the Beijing site was unavailable for a short period on May 25 (panel (a)).**



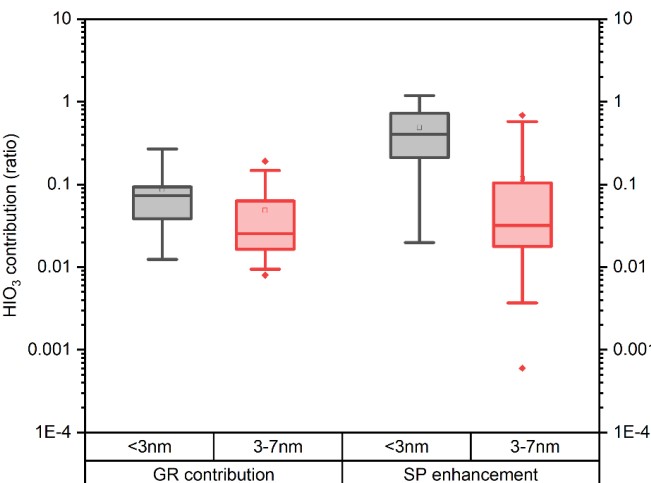

**Figure 7. The contributions in ratio of HIO$_3$ to growth rate (a) and survival probability enhancement (b) of particles within sub-3nm and 3-7nm in NPF events in the Beijing site utilizing the mode fitting method.**


**Table 1. Environmental factors and acid concentrations shown in Figure 5.**

| Site | Date | Start time | End time | H$_2$SO$_4$ (cm$^{-3}$) | HIO$_3$ (cm$^{-3}$) | CS (s$^{-1}$) | T (°C) | RH (%) | UVB/DSR (W/m$^2$) | O$_3$ (ppb) |
|---|---|---|---|---|---|---|---|---|---|---|
| Beijing | 2021/5/25 | 6:50:00 | 9:23:00 | 3.96E+06 | 2.03E+05 | 0.0046 | 18.5 | 26 | 0.20 | 25.68 |
| | 2021/5/26 | 9:20:00 | 10:48:36 | 8.01E+06 | 4.90E+05 | 0.0105 | 24.25 | 32 | 0.84 | 47.82 |
| | 2021/5/27 | 8:00:00 | 10:00:00 | 1.66E+06 | 5.41E+04 | 0.0222 | 19.6 | 59 | 0.45 | 11.17 |
| | 2021/5/28 | 6:00:00 | 8:15:00 | 5.35E+06 | 7.21E+05 | 0.0018 | 20.35 | 34 | 0.10 | 31.51 |
| | 2021/5/29 | 6:20:00 | 9:38:44 | 3.89E+06 | 4.99E+05 | 0.0040 | 19.4 | 50 | 0.25 | 31.24 |
| Nanjing | 2019/8/16 | 10:22:21 | 14:59:49 | 2.03E+07 | 1.40E+06 | 0.0200 | 33.23 | 40.69 | 858 | 105.38 |
| | 2019/8/17 | 8:04:22 | 11:10:40 | 4.27E+07 | 1.79E+06 | 0.0268 | 32.02 | 56.43 | 525 | 75.27 |
| | 2019/8/18 | 10:17:53 | 17:18:42 | 2.06E+07 | 2.20E+06 | 0.0134 | 30.93 | 39.52 | 705.1 | 81.09 |
| | 2019/8/19 | 8:00:00 | 10:00:00 | 1.35E+07 | 2.48E+06 | 0.0164 | 27.99 | 61.71 | 419.7 | 58.72 |
| | 2019/8/20 | 8:00:00 | 10:00:00 | 1.07E+07 | 2.12E+06 | 0.0154 | 27.4 | 67.08 | 458 | 48.26 |



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



*Data availability*. Measurement data at the AHL/BUCT and SORPES station, including acids concentration data, trace gas and aerosol data and meteorological data, are available upon request from the corresponding authors before the relevant databases are open to the public.


*Author contributions*. WN and XCH designed the research. YZ, CD, YG, YL, CH, TL and ZW conducted the measurements at the AHL/BUCT station. DL, YL, CL, LC, YL, LW and XC conducted the measurements at the SORPES station. YZ, DL, XCH, WN, CD, RC, YL, YG, TP, FB, XQ, PP, YL, CY, JJ, AD and MK analyzed the data and interpreted the results. YZ, DL, XCH and WN prepared the

manuscript with contributions from all co-authors.

*Competing interests*. The authors have no competing interests to declare.

*Acknowledgements*. We thank colleagues and students at the AHL/BUCT station and the SORPES

station for their contributions to the maintenance of the measurements.

Financial support. This work was supported by the National Natural Science Foundation of China (NSFC) project (92044301, 42220104006, 42075101 and 41975154), the Jiangsu Provincial Collaborative Innovation Center of Climate Change and the Fundamental Research Funds for the

Central Universities. Financial support from Samsung PM2.5 SRP is also acknowledged.