# Peer review of "Iodine oxoacids and their roles in sub-3 nanometer particle growth in polluted urban environments"

_EGUsphere, 2023_

## Referee Comment (RC2)

Review report on

**Iodine oxoacids and their roles in sub-3 nanometer particle growth in polluted urban environments**

By Zhang, Y., Li, D., He, X.-C., Nie, W., Deng, C., Cai, R., Liu, Y., Guo, Y., Liu, C., Li, Y., Chen, L., Li, Y., Hua, C., Liu, T., Wang, Z., Wang, L., Petäjä, T., Bianchi, F., Qi, X., Chi, X., Paasonen, P., Liu, Y., Yan, C., Jiang, J., Ding, A., and Kulmala, M.

The manuscript of Zhang et al. provides significant measurements of $HIO_x$ and their findings show the relation between new particle formation (NPF) and iodine oxoacids, in two polluted urban areas in China. Based on back trajectory analysis, it was found that iodine species mainly originate from marine and terrestrial sources instead of local human activity. Moreover, they calculated the contribution of $HIO_3$ and $H_2SO_4$ to growth rate and survival probability, when an NPF event occurred at both examined sites. Their findings indicate that during a NPF day $HIO_3$ promotes the survival of particles with diameters below 3 nm, while no impact is observed for particles between 3 and 7 nm. This study shines light into the crucial role of $HIO_3$ to NPF events under polluted conditions, using experimental measurements.

The manuscript is well written and interesting, with an added value of the presented results being from an area of the globe with significant population growth. However, there are several details missing and more thorough discussion should be made in specific sections. Other than that the paper can be recommended for publication after addressing the issues listed below.

1) L155-156: The authors consider that an undefined event is regarded as non-NPF event. Please comment. Also, do the authors believe that no freshly nucleated particles at the size below 3 nm is observed?

2) L302: Authors mention that the $H_2SO_4$ is lower in cold seasons. Could you elaborate on this?

3) L305: Please make a discussion about the calibration issues on $HIO_x$ measurements. What is the kind of calibration and the frequency?

4) L311: "The results indicate that the $H_2SO_4$ concentrations are generally higher than that of iodine oxoacids at both sites." -> Overall?? Because the time period of the measurements was different at both examined sites. What is the percentage difference among the $H_2SO_4$ iodine oxoacids concentrations at both sites? Why $H_2SO_4$ is constantly higher from iodine oxoacids. Please comment.

5) L155-156: The authors support that $H_2SO_4$ is the main contributor to NPF, however in Figure 1 increased $H_2SO_4$ concentrations are related to low NPF frequency. Please provide an explanation of why this feature does not occur here.

6) L356-357: "H$_2$SO$_4$ concentrations are higher in the spring and autumn, lower in the summer, and lowest in the winter." -> Please add an explanation regarding this seasonal diversity.

7) L400: "The authors found that ambient level of O$_3$ was not the limiting factor for HIO$_3$ formation." -> How the authors came to this conclusion? Could you elaborate on this? A reference is needed here.

8) L401: "…indicating that O$_3$ may influence terrestrial HIO$_3$ formation." -> It would be useful to provide a scatter plot for O$_3$ vs. HIO$_3$ to advocate this conclusion.

9) L403: "This mechanism mainly explains…" -> In which exactly process the authors refer to? Please explain.

10) L406: "Another possibility is that air temperature may strongly perturb the formation of HIO$_3$ …" -> A reference is needed here. This means that the augmentation of temperature involves increased HIO$_3$ concentrations? I would be careful of using the verb "perturb" here.

11) L428-429: "…land indicates that the continental outflows may also play a significant role in transporting HIO3 precursors." -> The authors consider that air masses coming from South have continental properties. However, in Figure S4a the southern cluster can travel over the sea, showing marine properties. Please comment on this?

12) L434: "…impact from residential coal burning and fossil fuel combustion power plant in Beijing is the largest." -> Are there any BC measurements in AHL/ BUCT station? A figure for BC vs. HIO$_3$ would be useful to advocates the author's conclusion.

13) L559: "Only events with clear growth…" -> This assumption means NPF events of class I. However, the classification of NPF episodes is not easily identifiable from the text. A classification could be added to the main text (2.2.1) and in the respective Table.

14) L567: "…environments with higher CoagS, such as Beijing, a fixed GR enhancement can lead to larger SP…" -> What does fixed GR enhancement mean? Please clarify.

15) L568-569: "Results in both Fig. 7 and Fig. S5 show that the median contribution of HIO$_3$ to the GR of particles in the 1.5-3 nm size range is 7.4% using the MOD method…" -> This contribution is referring to all NPF episodes in Beijing?

16) L569-571: "…whereas the contribution is only around 3% and 2% using the APT-x and APT-y methods, respectively. This is resulted from the difference in the measured GR calculated using either the APT or the MOD methods." -> Measured or calculated GR?? Therefore, this variation derives from the different estimated GR. What is the amount of uncertainty?

Technical corrections:

L24: "New particle formation processes contribute" -> New particle formation contributes

L25: "… ultrafine particles (UFP), and have great…" -> ultrafine particles (UFP; $d \leq 100$ nm)

L27: "…proved to dominate NPF events at some sites." -> proved to dominate NPF.

L28: "… we conducted a long-term comprehensive observation of gaseous…" -> … long-term measurements of gaseous …

L30: "… concentration in urban…" -> … concentration in both urban….

L31: "$HIO_3$ concentration is…"  -> $HIO_3$ is ….

L32-33: " …and is lowest in winter…" ->  … and is lowest in winter by xxx% and xxx%, respectively. $HIO_3$ exhibits more prominent variation than $H_2SO_4$ in both urban sites.

L34: "…temperature, radiation and ozone…" -> …temperature, solar radiation and ozone…

L40: "…suggesting HIOx are non-negligible contributor to UFPs in polluted urban areas." -> …suggesting that HIOx are significant contributor to UFPs in polluted urban areas.

L46-49: "Primary aerosol emissions include natural sources including the emission of sea spray, release of soil mineral dust, emission of biomass burning smoke, and the injection of volcanic debris (Claudio Tomasi, 2017) and anthropogenic emissions such as fuel combustion, industrial processes and transportation (Claudio Tomasi, 2017)." -> Primary aerosol emissions stem from natural sources, including sea spray, soil mineral dust, biomass burning, and volcanic debris (Claudio Tomasi, 2017), and anthropogenic sources such as fuel combustion, industrial processes and transportation (Claudio Tomasi, 2017).

L53: "…form new particles under atmospheric conditions…" -> … form new particles under appropriate atmospheric conditions…

L57: "…influence cloud formation and have climatic effects (Kerminen et al., 2005)." -> Use a more recent reference, e.g. Kalkavouras et al. (2019); Jiang et al. (2021)

L59: "…These small particles (< 50 nm) can penetrate…" -> You mean UFP, thus use $\leq 100$ nm instead of 50 nm in the parenthesis.

L60: "…understanding NPF processes is…" -> delete the word "processes"

L61: "…terms of predicting climate change…" -> Use evaluating instead of predicting

L70: "…vapours depending on the particle sizes. In urban Beijing,…" -> …vapours depending on the particle size. In Beijing…

L76-77: Use capital letter for O' Dowd.

L85: "…that $HIO_3$ (with $HIO_2$)…" -> Please clarify the content in the parenthesis. Its vague

L98: "…in urban Beijing from 2019 to 2021,…" -> …in urban Beijing from January 2019 to October 2021, …

L107-108: "The measurement in urban Beijing was conducted from January 2019 to October 2021. The site locates on the fifth floor…" -> Measurements in urban Beijing were conducted from January 2019 to October 2021, on the fifth floor…

L107-115: It would be useful to provide a figure with the exact location of both stations in main text or in the SM.

L122 and L124: Be sure about the unit of measurement

L125: "…from January 2020 to February 2020." -> or …from March 2020 to February 2020. ?

L135-142: Long sentence, please rephrase

L168: "…correction factor for mass flux as a function…" -> … correction factor for mass flux (Fuchs and Sutugin, 1970) as a function…

L174: "To quantify if notable growth is to occur, especially at sizes below a few nanometers, it is…" -> What do you mean a few nanometers? Please explain.

L225: "The counterbalance of CoagS and GR considerably affects the survival of small clusters." -> Please add a reference here.

L269: Put the sentence in line 269, after the equation (11).

L286: "Based on current knowledge about HIO3 formation pathways,…" -> Please add a reference here.

L315: "…from none to more than three quarters of the days…" -> Why present it as ¾ and not e.g. 75% ?

L347: "To better understand the roles of the studied acids in new particle formation and growth, we further…" -> To better understand the roles of the studied acids in NPF, we further….

L358: "The $HIO_3$ concentrations measured at the two sites are significantly lower than that at pristine coastal…" -> How low? Please add a percentage to express the variation of $HIO_3$.

L363: "… concentrations in Beijing and Nanjing are comparable to that in Helsinki, Finland." -> A reference is needed here.

L387-388: "The distinct diurnal variation in $HIO_x$ concentration with around one order of magnitude implies fast in-situ chemistry." -> This clear diurnal variation is observed during summer? It's vague. Please clarify.

L390: "This phenomenon is pronounced during summer daytime when daily maximum of $H_2SO_4$ appears around 2 hours earlier than that of $HIO_3$" -> This behavior was also apparent in autumn. It would be nice to refer this with some comments.

L393: "…cycle of $H_2SO_4$ follow that of radiation in spring, summer and winter." -> But not for fall? From Fig. 2 it is clear that the daily pattern of $H_2SO_4$ follows that of solar radiation.

L429-430: "…and terrestrial precursors (Li et al., 2014; Wang et al., 2017) may…" -> Terrestrial precursors such as?

L472: "…terrestrial biomes (Sive et al., 2007), and minor wetlands (Dimmer et al., 2001), biomass burning…" -> …terrestrial biomes (Sive et al., 2007), minor wetlands (Dimmer et al., 2001), and biomass burning…

L525-526: "When P is below 50 in clean environment or 100 in polluted urban cities, the SP of the sub-3 nm particles is agreed with the atmospheric observations." -> A reference is needed here.

L526: "As shown in Fig. S6, the…" -> The Fig. S6 in which station is referred to? The right y axis expresses the P? It is not clear.

L535-537: "Under the typical CoagS (around 0.0025 $s^{-1}$) at both sites, the SP could be enhanced by more than two orders of magnitude when GR is varied from 1 to 10 nm $h^{-1}$." -> It would be useful to provide a new figure, showing this CoagS value, because it's difficult for a reader to find/ see the value of 0.0025 $s^{-1}$.

L537-538: "Increased GR caused by additional condensing vapours enables faster growth, which in turn facilitates the survival of sub-10 nm particles from coagulation scavenging." -> Please add a reference here.

L540: "A limited variation of GR was suggested to cause considerable variation of SP (Cai et al., 2021a)." -> Repetition. The same sentence is written in the beginning of paragraph (line: 531-532).

L562-563: "The results in Table S2-S4 show that the contributions of $HIO_3$ to GR on May 25 and 26 were lower than 5%, whereas the contribution was more than 10% on May 29" -> In Table S4 the contribution of $HIO_3$ to GR for <3 nm particles is 9.4% (i.e. below 10%) on May 29. Furthermore, in this paragraph (L: 555-574) the authors analyze the MOD method. Why they add Tables S2 and S3?

L598-599: "At the SORPES station from June to November 2019, $HIO_3$ contributes 6.1% (median) and 6.7% (mean) to sub-3 nm particles growth compared to $H_2SO_4$." -> You mean that $HIO_3$ exhibits higher contributions compared to $H_2SO_4$ ?

Fig. S9: In the caption SORPES instead of SOREPS.

References

Kalkavouras, P., Bougiatioti, A., Kalivitis, N., Stavroulas, I., Tombrou, M., Nenes, A., and Mihalopoulos, N.: Regional new particle formation as modulators of cloud condensation nuclei

and cloud droplet number in the eastern Mediterranean, Atmos. Chem. Phys., 19, 6185–6203, https://doi.org/10.5194/acp-19-6185-2019, 2019

Jiang, S., Zhang, F., Ren, J., Chen, L., Yan, X., Liu, J., Sun, Y., and Li, Z.: Evaluation of the contribution of new particle formation to cloud droplet number concentration in the urban atmosphere, Atmos. Chem. Phys., 21, 14293–14308, https://doi.org/10.5194/acp-21-14293-2021, 2021.

Fuchs and Sutugin, Highly dispersed aerosol, Ann Arbour Science Publishers, Ann Arbour, London, 1970, https://doi.org/10.1016/B978-0-08-016674-2.50006-6, 1970.

---

## Author Comment (AC1)

**Response to the Anonymous Referee of the manuscript "Iodine oxoacids and their roles in sub-3 nanometer particle growth in polluted urban environments"**

Response to Anonymous Referee #1:

*This study reports unique long-term CIMS measurements of $HIO_x$ as well as NPF events in two urban areas, Beijing and Nanjing, China. The source analysis of iodic acid indicated potential long-range transport from marine and terrestrial sources rather than local anthropogenic emissions. They also estimated the contribution of $HIO_x$ and $H_2SO_4$ to nanoparticle growth rate and survival probability of nanoparticles. This paper shows that, although iodic acid does not play a major role in nanoparticle growth rate, the additional growth by iodic acid can enhance the survival of sub-3 nm particles in Beijing during NPF events. This suggests iodic acid contributes to NPF events in urban conditions with support from field measurements. However, some technical details on the measurement and analysis need to be clarified. Therefore, I recommend the manuscript for publication after revisions to address my comments below.*

We thank the Reviewer for carefully checking our manuscript and giving us helpful and constructive comments pointing us towards some important improvements in our analysis. Below we give a point-by-point response including the changes we made to the revised version of the manuscript. Comments are shown as *blue italic text* followed by our responses. Changes made to the revised manuscript are ==highlighted== and shown as "quoted underlined text" in our responses.

**General revision:**

We have renumbered all the supplementary figures according to their orders of appearance in the main text.

*(1) Eq. (9) & (11): What was the range of temperature/size correction factor applied to $GR(HIO_3)$ in eq. (9)? Also, please explain why $GR(H_2SO_4)$ in eq. (11) does not need temperature correction. In eq. (5), GR is corrected only to account for growth by condensation. And it is unclear in the text whether $GR(HIO_3)$ and $GR(H_2SO_4)$ in those equations are growth rates only by condensation (excluding coagulation).*

**Response:** Thanks for the comment. For $GR(HIO_3)$ calculation, which denotes the growth rates contributed by iodic acid ($HIO_3$), we adopted the CLOUD chamber experiment results obtained by

He et al. (2021). The growth rate measurements were carried out at 10 °C for the growth of 1.8-3.2 nm particles. Therefore, we corrected the fitted GR for both the temperature and size range difference using the Eq. (10). As depicted in Fig. R1, the correction factor for GR(HIO₃) is rather small (close to one) to both size range selection and temperature variation. The correction factor applied to field measurements at two sites and CLOUD chamber are close to 1. We further tested a larger temperature range from 10 to 50 °C in Fig. R1(b) and investigated the correction factors for the two size ranges studied here. The correction factors stay robust which suggests that the temperature and size range differences between the CLOUD experiments and field observations remain minimal. To further elaborate the calculations, the manuscript has been revised as below,

[Line 256-257] "To better understand the role of HIO₃ as an additional GR contributor at two sites, we calculate the condensational GR of HIO₃ and H₂SO₄ during NPF events, the computation criteria are listed as below."

[Line 277-281] "The correction factors remained consistently close to 1 (as shown in Fig. S2 for more detailed information). This suggests that the variations in size range and temperature between the CLOUD measurements and our field observations are minimal."

[Line 292-294] "H₂SO₄ contribution to GR is calculated as a first-order approximation independent of temperature as Eq. (11) (Stolzenburg et al., 2020), where $d_p = \dfrac{d_{p_{initial}} + d_{p_{final}}}{2}$ in nm, and the subscripts initial and final refer to the particle diameter at the beginning and the end of the growing process."

[Figure]

**Figure R1. The calculated correction factors based on Eq. (10) and the sensitivity against temperature and size. (a) The correction factors derived based on chamber and field site measurement conditions during NPF events. (b) The correction factor as a function of temperature with blue circles and grey squares denoting 1.5-3 nm and 3-7 nm growth, respectively. Vertical green line indicates the temperature of the CLOUD experiments (10 °C) and the yellow shadow indicates the temperature range of field observations.**

*(2) Line 273-274: "We define SP_tot as the particle survival probability calculated using the measured GRs (in Beijing) or the expected growth rate considering growth contributions from both H₂SO₄ and HIO₃ (in Nanjing)." Please explain how using different inputs for SP_tot calculation would affect EF in each site.*

**Response:** Thanks for the comment. We adopt different procedures here to derive the enhancement factor (EF) of survival probability (SP) because the growth trajectory of sub-3 nm particles is difficult to capture at SORPES due to the limitation of instruments. Furthermore, considering that $H_2SO_4$ is the dominant vapor for sub-3 nm range growth in urban environments (Deng et al., 2020a), we then let $SP_{tot}$ at SORPES equal to the SP considering both contributions from $H_2SO_4$ and $HIO_3$. Therefore, Eq. (12) for SORPES scenarios can be clarified as

$$\text{EF} = \frac{SP\left(GR_{H_2SO_4} + GR_{HIO_3}\right)}{SP\left(GR_{H_2SO_4}\right)} - 1. \tag{R1}$$

As shown in Fig. S13, the amplification of SP is especially sensitive against initial GR ($GR_{H_2SO_4}$ in this case). The $GR_{H_2SO_4}$ and $GR_{HIO_3}$ at SORPES is derived based on the empirical equation and therefore, the measurement of two acids will introduce uncertainties for EF calculation at SORPES. On the contrary, the biggest uncertainty for EF calculation at BUCT/AHL will be the fitting process for measured GR in NPF events for sub-3 nm particle growth. Besides, the EF for 3-7 nm range growth is also calculated at BUCT/AHL but not at SORPES.

The values of EF for sub-3 nm particle growth at both sites can be checked via Table S7 and Table S9.

*(3) Line 305-306: "The calibrated HIO_x concentration is above the detection limit during almost the entire measurement periods…" However, I can not find more details on the calibration, except for the general description of nitrate CIMS in section 2.1.2. Therefore, the quantification method of HIO_x (as well as H₂SO₄) needs to be shown and the sources of uncertainty need to be discussed since HIO_x concentrations are being used for the growth rate and survival probability calculations. How and how often were the instruments calibrated for HIO_x? Do authors expect losses in the sampling line or variations of HIO_x sensitivity due to changes in temperature/relative humidity?*

**Response:** Thanks for the suggestions. We have now included a brief discussion of the calibration processes, the references and the expected systematic error of our calibration method. And the manuscript has been revised as below,

**[Line 135-146]** "H₂SO₄ calibration was conducted using a standardized method (Kurten et al., 2012; He et al., 2023). In a nutshell, the calibration of H₂SO₄ involved the reaction of an excessive amount of sulfur dioxide (SO₂) with a known quantity of hydroxyl (OH) radicals generated by a portable

mercury lamp. This mercury lamp is equipped with a filter to intercept the sample air containing water, which, in turn, is photolyzed to produce OH radicals. The convection-diffusion-reaction processes within the chemical ionization inlet can be accurately simulated using a two-dimensional model (e.g., the MARFORCE-Flowtube model) (He et al., 2023), allowing for the quantification of $H_2SO_4$ concentration at the mass spectrometer's entrance. The quantification of the measured signals for $H_2SO_4$ monomer at both sites are seasonally calibrated with diffusion losses in the inlet tube taken into consideration. Since both $H_2SO_4$ and $HIO_x$ are detected at the collision limit, they share the same calibration factor (He et al., 2021; He et al., 2023). The general systematic error for the detection of $H_2SO_4$ and $HIO_x$ is expected to be within 50% to 200% for field observations (Liu et al., 2021)."

*(4) Line 426: "implies that marine iodine sources could be important" Could it be emissions from industrial areas along with the back trajectory path?*

**Response:** Thanks for pointing out potential iodine sources. We use the back-trajectory analysis in this study to investigate the potential sources of $HIO_3$ precursor because of the limited knowledge of their sources in inland cities. Marine environments emit both inorganic iodine through ozone-stimulated liberation of HOI and $I_2$ from surface ocean and organic iodine compounds through biotic and abiotic processes such as $CH_3I$ and $CH_2I_2$. In general, the inorganic iodine molecules, HOI and $I_2$, dominate the global iodine sources (Carpenter et al., 2013; Ordóñez et al., 2012). The measurement site of BUCT/AHL is 150 km away from the nearest coastline on the southeast side, therefore, we link the trajectory cluster from the same direction with marine sources, which accounts for the majority of level 1 clusters.

As for the industrial sources of iodine, a previous work conducted in urban Beijing found that intensive coal combustions are sources of atmospheric iodine during heating seasons (Shi et al., 2021), in the form of organic iodine compounds in northern China. However, the cluster analysis for back-trajectories at BUCT/AHL are specially for warm months (from May to September), so it can be inferred that the industrial emission of iodine compounds is less significant compared to the dominant iodine source of oceans.

We have specified the time window for back-trajectory analysis in Beijing, and the modified manuscript is shown as below,

**[Line 458-460]** "In order to investigate the source of $HIO_x$ in urban environments, we further conduct cluster analysis of the air mass backward trajectories of the AHL/BUCT station in warm seasons (from May to September in this study)."

*(5) Line 428: That is the opposite direction of continental outflow.*

**Response:** Thank you for your comment. The northern cluster instead of the southern one would indicate the continental outflow better. Therefore, we corrected the expression in manuscript as following, according to your reminder.

**[Line 465-466]** "Additionally, the air mass travels from northern wind may also carry substantial precursors of $HIO_3$, indicating the potential terrestrial sources of iodine."

*(6) Line 445: Even if the $HIO_3$ emission (or secondary formation) is higher in winter (when $PM_{2.5}$ loading is higher), still the concentration of $HIO_3$ can be lower due to potentially higher CS. Thus the y-axis in Fig.3 can be normalized by CS to see if the source of $HIO_3$ is correlated to $PM_{2.5}$ pollution.*

**Response:** Thank you for the good suggestions. Taken the steady-state assumption of ambient $HIO_3$, we get to evaluate the production of $HIO_3$ according to its loss rate, which is dominantly condensational loss. Therefore, the product of $HIO_3$ and CS is used to describe the $HIO_3$ production rate here and its correlation with $PM_{2.5}$ is shown as Fig. R2. The results in the following figure show that there is no apparent impact of $PM_{2.5}$ level on $HIO_3$ production. To avoid the seasonal variation of $PM_{2.5}$, we particularly investigate the correlation between $HIO_3$ production and $PM_{2.5}$ in warm seasons in Fig. R2(b), and there is no significant correlation, either. Additionally, as depicted in Fig. R3, the relationship of CS and $PM_{2.5}$ is further examined, which shows a significant positive relationship. Therefore, if $PM_{2.5}$ pollution does contribute to the production of $HIO_3$, we would have expected to see the accordingly higher values of $HIO_3$*CS, because of higher CS and assumingly higher $HIO_3$ concentration. However, as depicted in Fig. R2, under high level of $PM_{2.5}$ pollution, there is no significant elevation of $HIO_3$ production. To conclude, the production of $HIO_3$ is not correlated to $PM_{2.5}$ pollution based on our measurements.

The relationship between $HIO_3$ production rate and $PM_{2.5}$ have been mentioned in the revised manuscript as following,

**[Line 486-488]** "Furthermore, the results in Fig. S7 demonstrate that $PM_{2.5}$ pollution does not play a conclusive role on the $HIO_3$ production, especially with the seasonal influence excluded."

[Figure]

**Figure R2. The impact of PM$_{2.5}$ on HIO$_3$ production calculated from the HIO$_3$ concentration and CS (a) in all seasons and (b) specifically in warm seasons (from May to September).**

[Figure]

**Figure R3. The correlation between CS and PM$_{2.5}$.**

*(7) Line 522-529: Survival parameter (P) is newly introduced here and it is unclear why P is being used for AHL/BUCT case while survival probability (SP) is used for the rest of the analysis. Please compare the survival parameter from Kulmala et al. (2017) and survival probability (SP) from Lehtinen et al. (2007) and explain why using P is useful here.*

**Response:** Thanks for your good comments. The primary role of the survival parameter $P$ (Kulmala et al., 2017) was to provide a viewpoint for the occurrence of NPF, similar to $L$ or $L_\Gamma$ (Mcmurry et al., 2005; Kuang et al., 2010; Cai et al., 2017), especially in urban environments. Conversely, survival probability (SP) serves as a quantitative estimate of what is the percentage of the recently formed particles that grows to larger sizes.

The survival parameter $P$ was used here to characterize the particle formation events. Events with $P$ smaller than 100 were selected in this study as the particle formation and growth under low CS conditions are more certain (Kulmala et al., 2017). When the $P$ is larger than 100, the mechanisms of particle formation and growth under high CS conditions is still under debate (Kulmala et al., 2017) and thus these data are avoided.

Given that measured GRs are not derived due to the absence of sub-3 nm particle measurement, the survival parameter P is not calculated for SORPES station.

*(8) Line 567: "a fixed GR enhancement" What does "fixed" mean?*

**Response:** Our apologies for not being clear. In this section, we were trying to discuss about the sensitivity of SP enhancement against GR enhancement, which is controlled by CoagS.

For Eq. (8), by considering that $d_{p1}$, $d_{p2}$ and CoagS are constant, we can further combine the constant values besides CoagS as a new term A.

Where

$$\text{A} = \frac{d_{p1}}{m-1}\left[\left(\frac{d_{p2}}{d_{p1}}\right)^{1-m} - 1\right], \tag{R2}$$

and Eq. (8) can be simplified as

$$\text{SP} = \exp\left(\frac{CoagS}{A} \cdot \frac{1}{GR}\right). \tag{R3}$$

By assuming m=1.7, it can be deduced that A<0, and if we take the derivative of SP with respect to GR for Eq. (R3), we further get the following Eq. (R4):

$$\frac{\Delta SP}{\Delta GR} = -\frac{1}{GR^2} \cdot \frac{CoagS}{A} \cdot SP. \tag{R4}$$

For a given GR value and GR enhancement ($\Delta$GR), the enhancement of SP ($\Delta$SP) is subject to the value of CoagS. The larger the CoagS is, the larger enhancement of SP can be expected.

For the two cases studied in this section, the GR values are similar while on Jun 21, 2021, the CoagS is 5 times larger than May 29, 2021, leading to a more pronounced enhancement of SP on the former day. The expressions we were using was somehow confusing, and we have revised the statement as below,

**[Line 608-609]** "This suggests that in polluted environments with higher CoagS, such as Beijing,

SP enhancement can be more sensitive to GR enhancement.”

*(9) Line 570-575: Are the MOD, APT-x, and APT-y methods equally credible? Then the differences in those methods represent the uncertainty of using MOD fitting method? Or do APT-x and APT-y represent upper/lower limit of MOD fitting?*

**Response:** Thank you for your comments. We acknowledge the need for clearer articulation regarding the methods used for GR calculations. To provide a more comprehensive explanation, we have expanded upon these methods in the revised supplemental information in **[Line 1296-1317]**.

S5. The calculation method of GRs in NPF events

The growth of newly formed particles for NPF events are often reflected by the collective shift of measured particle size distribution towards larger sizes as time evolves, but it is unfeasible to track the growth of a single particle based on the measurements. Therefore, both approaches used in this study (mode-fitting and appearance time) are referred to as collective approaches and the GRs in this study are the estimated ones (Stolzenburg et al., 2023). While using mode-fitting method, the growth trajectory of new particles is represented by the peak diameter ($d_p$) of the nucleation mode, which is determined after applying log-normal distributions to the measured size distribution (Kulmala et al., 2012). For measured particle number size distribution at each time ($t$), there will be a $d_p$, and the value of GR ($GR_{mode}$) is derived by a linear fit to the $d_p$ vs $t$.

Instead of tracking the shift of peak diameter for a given time period, appearance time method seeks to find the time it takes ($\Delta t$) for the particle to grow between instrument size bins ($\Delta d_p$). In this study, we take the time ($t$) that the measured concentration of particles reaches its half maximum for each $d_p$. For each particle size bin ($d_p$) of the instrument used at BUCT/AHL, there will be a $t$, and the value of GR ($GR_{apt}$) is derived by a linear fit to the $d_p$ vs $t$. Additionally, we believe it would be conceptually more correct to consider diameter as the independent variable when fitting the GRs using appearance time method although previous studies commonly take appearance time as the independent variable and diameter as the dependent variable when fitting the GRs. That is because each data point corresponds to a precise size bin, and any variation (largely stemming from uncertainty due to atmospheric heterogeneity) among the data points primarily exists in appearance time. Consequently, the fitting method with diameter as the independent variable was named as APT-y and the widely used one with time as the independent variable was called as APT-x.

However, it is important to note that the ongoing debate regarding the accuracy between the MOD and APT methods remains unresolved. At present, there is not a definitive conclusion regarding which method is more credible. Both the MOD (mode-fitting) and the APT (appearance time) methods are commonly employed in determining measured GR in NPF events (Kulmala et al., 2012). Recent studies have shown a prevalent use of the MOD method in calculating GR and recognition of particle growth contributor (i.e., $H_2SO_4$ and OOMs) (Deng et al., 2020b; Qiao et al.,

2021). Hence, in this study's main text, we have utilized the MOD method for comparison with previous research. However, the results from APT-x and APT-y methods are shown in the SI and are reflected in the main text. Hence, in this study's main text, we have utilized the MOD method for comparison with previous research. However, the results from APT-x and APT-y methods are shown in the SI and are reflected in the main text.

It is imperative to view the discrepancies among these methodologies as inherent uncertainties in GR calculations rather than solely attributed to the MOD method itself. For instance, the GRs for sub-3 nm particles determined using APT-x, APT-y, and MODE on May 26, 2021, were 3.17, 5.13, and 2.22 nm h$^{-1}$, respectively, resulting in an average discrepancy of 1.93 nm h$^{-1}$ between both APT methods and MODE. However, these values changed to 1.50, 1.92, and 1.93 nm h$^{-1}$, respectively, on Jun 21, 2021, leading to a comparatively minor deviation. Moreover, the disparity in GRs for 3-7 nm particles across these methods is more substantial as expected, given that the GRs for these particles are consistently more than those of sub-3 nm particles. Our analysis reveals an average discrepancy among those methods of 1.5 nm h$^{-1}$ for sub-3 nm particles and 3.1 nm h$^{-1}$ for 3-7 nm particles, reflecting the level of uncertainty associated with GR determination.

*(10) Line 586-588: "This is based on the observed consistency between the gaseous H$_2$SO$_4$ concentration and its significant contribution to the sub-3 nm particle growth rate in Beijing (Deng et al., 2020b)." However, in this study (Table S4), GR$_{1.5-3}$ was generally higher than GR(H$_2$SO$_4$) in Beijing indicating contributions from other species.*

**Response:** Thanks for the comment. Sulfuric acid (H$_2$SO$_4$, SA) is one of the most important species in atmospheric nucleation and significantly contributes to particle initial growth (Dunne et al., 2016; Yao et al., 2018; Deng et al., 2020a). The GR(H$_2$SO$_4$) is the estimated contribution of SA to particle condensational growth according to Stolzenburg et al. (2020). However, the reviewer is right that other organic materials may make some contribution to the initial growth as well (Salma and Németh, 2019; Mohr et al., 2019; Deng et al., 2020a). Therefore, with respect to our analysis, we remark SA as a significant GR contributor instead of the sole dominant vapor for particle growth in all sizes. Also, the results of SORPES regarding the discussion of SP enhancement are therefore disadvantaged compared to the BUCT/AHL results due to the lack of direct GR measurements. And the revised statements are shown as below,

**[Line 295-299]** "But it should be noted that H$_2$SO$_4$ does not solely dominate the initial growth process, other organic species may make some contribution as well (Stolzenburg et al., 2023). Therefore, in this study, the results calculated at SORPES are disadvantaged compared to the BUCT/AHL data using directly measured GRs. This part of the results should be considered a compromise due to the absence of direct GR measurements."

*(11) Fig. 5: In my understanding of the text, although HIO$_3$ is not the major contributor in the measured GR, the sub 3-nm SP is so sensitive to additional GR*

*from HIO₃ that the additional HIO₃ enhances SP of sub 3-nm significantly (as shown in fig. 7). And fig. 5 is a nice place to visually indicate that sensitive regimes that correspond to the data in fig. 7.*

**Response:** Thanks for your good suggestion. We have put the field data points in Figure 5 to visually indicate the sensitive regimes of SP to GR. From the following figure (Figure R4), the results demonstrate that minor HIO₃ contribution to GR can result in significant enhancement of particle survival probability, especially in the sensitivity transition regimes for sub-3 nm particles (from blue to red).

[Figure]

**Figure R4. The effect of coagulation sink (CoagS) and growth rate (GR) on particle survival probability for 1.5-3 nm (a) and 3-7 nm (b) particles, respectively. The triangular and circular points represent GR without and with iodic acid contribution, respectively.**

*(12) Line 606-608: "As depicted in Fig. S8(c), SP enhancements in percentage are generally one order of magnitude higher than the GR contribution in percentage and HIO₃ can result in as high as 2-fold enhancement on SP in sub-3 nm particle growth." How important is the ~100% (or ~40% on average) enhancement of SP*

*from HIO₃ in the occurrence of NPF event? In Table S4, during NPF events in Beijing, the survival probability ("P1") of sub 3-nm particles spans over 3 orders of magnitude from 6.1E-4 to 2.1E-1. Also, that wide range of SP needs to be directly mentioned in the main text.*

**Response:** Thanks for the suggestions. NPF is a crucial process for the Earth's aerosol system as it is estimated to contribute majorly to aerosol population as well as cloud condensation nuclei (CCN) (Kuang et al., 2009; Gordon et al., 2017). A population of newly formed particles need to grow efficiently to reach typical sizes (usually accumulation or Aitken-mode sizes) in order to be activated to become CCNs, during which they are likely to be scavenged by pre-existing aerosol particles (Mcfiggans et al., 2006; Kulmala et al., 2001). Hence, survival probability (SP), which characterizes the fraction of freshly formed particles that survive the scavenging process, is an essential and irreplaceable parameter for evaluating the atmospheric influence of NPF. The knowledge of particle SP in ambient atmosphere is limited despite its importance. In this study, we not only reported the theoretical SP calculated using GR and CoagS, and we estimate the influence the potential enhancement of SP with the help of additional GR contributor as well, such as HIO₃. The derived enhancement (up to 100% in some cases) indicate that the fraction of freshly nucleated particles survives the scavenging after a certain growth process is doubled, thus enhancing the atmospheric influences of NPF events. Furthermore, the measurement sites in this study are both located in environments with intensive human impacts where the coagulation sinks are commonly higher. Therefore, the enhancement of SP is even more significant for the new particles to survive the counterbalance of growth process and scavenging by pre-existing aerosols.

Following your suggestions, we described the wide range of SP variations in the revised manuscript.

**[Line 573-575]:** "It also means that the SP of these newly formed particles exhibits a substantial variability, spanning more than three orders of magnitude, as illustrated in Tables S2~4."

**Technical comments:**

*(1) Line 94: "survival probability (Kulmala et al., 2017)..." According to the rest of the text, the survival parameter is from Kulmala et al. (2017) and the survival probability is from Lehtinen et al. (2007).*

**Response:** Thanks for the suggestion. We have replaced the reference here.

*(2) Line 250: GR'(HIO3) -> GR(HIO3).*

**Response:** Corrected.

*(3) Line 269: Define $dp_{initial}$ and $dp_{final}$.*

**Response:** Thanks for the suggestion. We have revised the statement here to be more specific:

**[Line 299]** "where $d_p = \frac{d_{p\,initial} + d_{p\,final}}{2}$ in nm, and the subscripts initial and final refer to the particle diameter at the beginning and the end of the growing process. [$H_2SO_4$] is the gas phase $H_2SO_4$ concentration in molecule cm$^{-3}$."

*(4) Line 594: "IA" and "SA" not defined previously (supposedly HIO3 and H2SO4?). Use consistent names in the main text and SI unless otherwise needed.*

**Response:** We apologize for not being clear. We have checked the relevant expressions to give the consistent names for acids.

*(5) Fig. 1&6: Make the panels bigger.*

Response: Updated.

*(6) Table S8 and S9 caption typo: SOREPS -> SORPES.*

**Response:** Corrected.

*(6) Table S2-S7 & S9: Please use the consistent notations for survival probability (SP) and enhancement factor (EF) rather than "P" and "E". Especially, "P" was used for the survival parameter in the main text and Fig. S6.*

**Response:** Corrected.

*(7) Fig S4: The back trajectory color scheme is not consistent. Also, specify the meaning of (%) that the colors represent in the caption.*

**Response:** Thanks for the suggestion. Actually, the color scheme used for the back trajectories is consistent with their respective levels across all four panels. Furthermore, the percentage associated with each trajectory indicates the probability of air mass originating from the corresponding cluster. Also, we have added some detailed discussions about the analysis in the caption of Fig. S5.

And to avoid confusion caused by the color of different clusters, we further revise Fig. S5 as below,

[Figure]

**Figure S5. The cluster analysis in different HIO$_3$ precursors intensities.** The four levels of the proxy concentration of HIO$_3$ precursors are in the 75% - 100%, 50%-75%, 25%-50%, 0-25% percentiles from the first to the fourth levels, respectively. The percentage of each trajectory reflects the ratio of the corresponding cluster.

**References**

Cai, R., Yang, D., Fu, Y., Wang, X., Li, X., Ma, Y., Hao, J., Zheng, J., and Jiang, J.: Aerosol surface area concentration: a governing factor in new particle formation in Beijing, ACP, 17, 12327-12340, 10.5194/acp-17-12327-2017, 2017.

Carpenter, L. J., MacDonald, S. M., Shaw, M. D., Kumar, R., Saunders, R. W., Parthipan, R., Wilson, J., and Plane, J. M. C.: Atmospheric iodine levels influenced by sea surface emissions of inorganic iodine, Nature Geoscience, 6, 108-111, 10.1038/ngeo1687, 2013.

Deng, C., Fu, Y., Dada, L., Yan, C., Cai, R., Yang, D., Zhou, Y., Yin, R., Lu, Y., Li, X., Qiao, X., Fan, X., Nie, W., Kontkanen, J., Kangasluoma, J., Chu, B., Ding, A., Kerminen, V.-M., Paasonen, P., Worsnop, D. R., Bianchi, F., Liu, Y., Zheng, J., Wang, L., Kulmala, M., and Jiang, J.: Seasonal Characteristics of New Particle Formation and Growth in Urban Beijing, Environmental Science & Technology, 54, 8547-8557, 10.1021/acs.est.0c00808, 2020a.

Deng, C., Fu, Y., Dada, L., Yan, C., Cai, R., Yang, D., Zhou, Y., Yin, R., Lu, Y., Li, X., Qiao, X., Fan, X., Nie, W., Kontkanen, J., Kangasluoma, J., Chu, B., Ding, A., Kerminen, V. M., Paasonen, P., Worsnop, D. R., Bianchi, F., Liu, Y., Zheng, J., Wang, L., Kulmala, M., and Jiang, J.: Seasonal Characteristics of New Particle Formation and Growth in Urban Beijing, EST, 54, 8547-8557, 10.1021/acs.est.0c00808, 2020b.

Dunne, E. M., Gordon, H., Kürten, A., Almeida, J., Duplissy, J., Williamson, C., Ortega, I. K., Pringle, K. J., Adamov, A., Baltensperger, U., Barmet, P., Benduhn, F., Bianchi, F., Breitenlechner, M., Clarke, A., Curtius, J., Dommen, J., Donahue, N. M., Ehrhart, S., Flagan, R. C., Franchin, A., Guida, R., Hakala, J., Hansel, A., Heinritzi, M., Jokinen, T., Kangasluoma, J., Kirkby, J., Kulmala, M., Kupc, A., Lawler, M. J., Lehtipalo, K., Makhmutov, V., Mann, G., Mathot, S., Merikanto, J., Miettinen, P., Nenes, A., Onnela, A., Rap, A., Reddington, C. L. S., Riccobono, F., Richards, N. A. D., Rissanen, M. P., Rondo, L., Sarnela, N., Schobesberger, S., Sengupta, K., Simon, M., Sipilä, M., Smith, J. N., Stozkhov, Y., Tomé, A., Tröstl, J., Wagner, P. E., Wimmer, D., Winkler, P. M., Worsnop, D. R., and Carslaw, K. S.: Global atmospheric particle formation from CERN CLOUD measurements, Science, 354, 1119-1124, doi:10.1126/science.aaf2649, 2016.

Gordon, H., Kirkby, J., Baltensperger, U., Bianchi, F., Breitenlechner, M., Curtius, J., Dias, A., Dommen, J., Donahue, N. M., Dunne, E. M., Duplissy, J., Ehrhart, S., Flagan, R. C., Frege, C., Fuchs, C., Hansel, A., Hoyle, C. R., Kulmala, M., Kürten, A., Lehtipalo, K., Makhmutov, V., Molteni, U., Rissanen, M. P., Stozkhov, Y., Tröstl, J., Tsagkogeorgas, G., Wagner, R., Williamson, C., Wimmer, D., Winkler, P. M., Yan, C., and Carslaw, K. S.: Causes and importance of new particle formation in the present-day and preindustrial atmospheres, Journal of Geophysical Research: Atmospheres, 122, 8739-8760, 10.1002/2017jd026844, 2017.

He, X.-C., Tham, Y. J., Dada, L., Wang, M., Finkenzeller, H., Stolzenburg, D., Iyer, S., Simon, M., Kürten, A., Shen, J., Rörup, B., Rissanen, M., Schobesberger, S., Baalbaki, R., Wang, D. S., Koenig, T. K.,

Jokinen, T., Sarnela, N., Beck, L. J., Almeida, J., Amanatidis, S., Amorim, A., Ataei, F., Baccarini, A., Bertozzi, B., Bianchi, F., Brilke, S., Caudillo, L., Chen, D., Chiu, R., Chu, B., Dias, A., Ding, A., Dommen, J., Duplissy, J., El Haddad, I., Gonzalez Carracedo, L., Granzin, M., Hansel, A., Heinritzi, M., Hofbauer, V., Junninen, H., Kangasluoma, J., Kemppainen, D., Kim, C., Kong, W., Krechmer, J. E., Kvashin, A., Laitinen, T., Lamkaddam, H., Lee, C. P., Lehtipalo, K., Leiminger, M., Li, Z., Makhmutov, V., Manninen, H. E., Marie, G., Marten, R., Mathot, S., Mauldin, R. L., Mentler, B., Möhler, O., Müller, T., Nie, W., Onnela, A., Petäjä, T., Pfeifer, J., Philippov, M., Ranjithkumar, A., Saiz-Lopez, A., Salma, I., Scholz, W., Schuchmann, S., Schulze, B., Steiner, G., Stozhkov, Y., Tauber, C., Tomé, A., Thakur, R. C., Väisänen, O., Vazquez-Pufleau, M., Wagner, A. C., Wang, Y., Weber, S. K., Winkler, P. M., Wu, Y., Xiao, M., Yan, C., Ye, Q., Ylisirniö, A., Zauner-Wieczorek, M., Zha, Q., Zhou, P., Flagan, R. C., Curtius, J., Baltensperger, U., Kulmala, M., Kerminen, V.-M., Kurtén, T., Donahue, N. M., Volkamer, R., Kirkby, J., Worsnop, D. R., and Sipilä, M.: Role of iodine oxoacids in atmospheric aerosol nucleation, Science, 371, 589-595, 10.1126/science.abe0298, 2021.

He, X. C., Shen, J., Iyer, S., Juuti, P., Zhang, J., Koirala, M., Kytökari, M. M., Worsnop, D. R., Rissanen, M., Kulmala, M., Maier, N. M., Mikkilä, J., Sipilä, M., and Kangasluoma, J.: Characterisation of gaseous iodine species detection using the multi-scheme chemical ionisation inlet 2 with bromide and nitrate chemical ionisation methods, Atmos. Meas. Tech., 16, 4461-4487, 10.5194/amt-16-4461-2023, 2023.

Kuang, C., McMurry, P. H., and McCormick, A. V.: Determination of cloud condensation nuclei production from measured new particle formation events, GRL, 36, 10.1029/2009gl037584, 2009.

Kuang, C., Riipinen, I., Sihto, S. L., Kulmala, M., McCormick, A. V., and McMurry, P. H.: An improved criterion for new particle formation in diverse atmospheric environments, ACP, 10, 8469-8480, 10.5194/acp-10-8469-2010, 2010.

Kulmala, M., Kerminen, V. M., Petaja, T., Ding, A. J., and Wang, L.: Atmospheric gas-to-particle conversion: why NPF events are observed in megacities?, Faraday Discuss, 200, 271-288, 10.1039/c6fd00257a, 2017.

Kulmala, M., Maso, M. D., Mäkelä, J. M., Pirjola, L., Väkevä, M., Aalto, P., Miikkulainen, P., Hämeri, K., and O'dowd, C. D.: On the formation, growth and composition of nucleation mode particles, Tellus B: Chemical and Physical Meteorology, 53, 479-490, 10.3402/tellusb.v53i4.16622, 2001.

Kulmala, M., Petäjä, T., Nieminen, T., Sipilä, M., Manninen, H. E., Lehtipalo, K., Dal Maso, M., Aalto, P. P., Junninen, H., Paasonen, P., Riipinen, I., Lehtinen, K. E. J., Laaksonen, A., and Kerminen, V.-M.: Measurement of the nucleation of atmospheric aerosol particles, Nature Protocols, 7, 1651-1667, 10.1038/nprot.2012.091, 2012.

Kurten, A., Rondo, L., Ehrhart, S., and Curtius, J.: Calibration of a Chemical Ionization Mass Spectrometer for the Measurement of Gaseous Sulfuric Acid, Journal of Physical Chemistry A, 116, 6375-6386, 10.1021/jp212123n, 2012.

Liu, Y., Nie, W., Li, Y., Ge, D., Liu, C., Xu, Z., Chen, L., Wang, T., Wang, L., Sun, P., Qi, X., Wang, J., Xu, Z., Yuan, J., Yan, C., Zhang, Y., Huang, D., Wang, Z., Donahue, N. M., Worsnop, D., Chi, X., Ehn, M., and Ding, A.: Formation of condensable organic vapors from anthropogenic and biogenic volatile organic compounds (VOCs) is strongly perturbed by NOx in eastern China, Atmos. Chem. Phys., 21, 14789-14814, 10.5194/acp-21-14789-2021, 2021.

McFiggans, G., Artaxo, P., Baltensperger, U., Coe, H., Facchini, M. C., Feingold, G., Fuzzi, S., Gysel, M., Laaksonen, A., Lohmann, U., Mentel, T. F., Murphy, D. M., O'Dowd, C. D., Snider, J. R., and Weingartner, E.: The effect of physical and chemical aerosol properties on warm cloud droplet activation, Atmos. Chem. Phys., 6, 2593-2649, 10.5194/acp-6-2593-2006, 2006.

McMurry, P. H., Fink, M., Sakurai, H., Stolzenburg, M. R., Mauldin, R. L., Smith, J., Eisele, F., Moore, K., Sjostedt, S., Tanner, D., Huey, L. G., Nowak, J. B., Edgerton, E., and Voisin, D.: A criterion for new particle formation in the sulfur-rich Atlanta atmosphere, Journal of Geophysical Research, 110, 10.1029/2005jd005901, 2005.

Mohr, C., Thornton, J. A., Heitto, A., Lopez-Hilfiker, F. D., Lutz, A., Riipinen, I., Hong, J., Donahue, N. M., Hallquist, M., Petäjä, T., Kulmala, M., and Yli-Juuti, T.: Molecular identification of organic vapors driving atmospheric nanoparticle growth, Nature Communications, 10, 4442, 10.1038/s41467-019-12473-2, 2019.

Ordóñez, C., Lamarque, J. F., Tilmes, S., Kinnison, D. E., Atlas, E. L., Blake, D. R., Sousa Santos, G., Brasseur, G., and Saiz-Lopez, A.: Bromine and iodine chemistry in a global chemistry-climate model: description and evaluation of very short-lived oceanic sources, Atmos. Chem. Phys., 12, 1423-1447, 10.5194/acp-12-1423-2012, 2012.

Qiao, X., Yan, C., Li, X., Guo, Y., Yin, R., Deng, C., Li, C., Nie, W., Wang, M., Cai, R., Huang, D., Wang, Z., Yao, L., Worsnop, D. R., Bianchi, F., Liu, Y., Donahue, N. M., Kulmala, M., and Jiang, J.: Contribution of Atmospheric Oxygenated Organic Compounds to Particle Growth in an Urban Environment, EST, 55, 13646-13656, 10.1021/acs.est.1c02095, 2021.

Salma, I. and Németh, Z.: Dynamic and timing properties of new aerosol particle formation and consecutive growth events, Atmos. Chem. Phys., 19, 5835-5852, 10.5194/acp-19-5835-2019, 2019.

Shi, X., Qiu, X., Chen, Q., Chen, S., Hu, M., Rudich, Y., and Zhu, T.: Organic Iodine Compounds in Fine Particulate Matter from a Continental Urban Region: Insights into Secondary Formation in the Atmosphere, Environ Sci Technol, 55, 1508-1514, 10.1021/acs.est.0c06703, 2021.

Stolzenburg, D., Cai, R., Blichner, S. M., Kontkanen, J., Zhou, P., Makkonen, R., Kerminen, V.-M., Kulmala, M., Riipinen, I., and Kangasluoma, J.: Atmospheric nanoparticle growth, Reviews of Modern Physics, 95, 045002, 10.1103/RevModPhys.95.045002, 2023.

Stolzenburg, D., Simon, M., Ranjithkumar, A., Kürten, A., Lehtipalo, K., Gordon, H., Ehrhart, S.,

Finkenzeller, H., Pichelstorfer, L., Nieminen, T., He, X.-C., Brilke, S., Xiao, M., Amorim, A., Baalbaki, R., Baccarini, A., Beck, L., Bräkling, S., Caudillo Murillo, L., Chen, D., Chu, B., Dada, L., Dias, A., Dommen, J., Duplissy, J., El Haddad, I., Fischer, L., Gonzalez Carracedo, L., Heinritzi, M., Kim, C., Koenig, T. K., Kong, W., Lamkaddam, H., Lee, C. P., Leiminger, M., Li, Z., Makhmutov, V., Manninen, H. E., Marie, G., Marten, R., Müller, T., Nie, W., Partoll, E., Petäjä, T., Pfeifer, J., Philippov, M., Rissanen, M. P., Rörup, B., Schobesberger, S., Schuchmann, S., Shen, J., Sipilä, M., Steiner, G., Stozhkov, Y., Tauber, C., Tham, Y. J., Tomé, A., Vazquez-Pufleau, M., Wagner, A. C., Wang, M., Wang, Y., Weber, S. K., Wimmer, D., Wlasits, P. J., Wu, Y., Ye, Q., Zauner-Wieczorek, M., Baltensperger, U., Carslaw, K. S., Curtius, J., Donahue, N. M., Flagan, R. C., Hansel, A., Kulmala, M., Lelieveld, J., Volkamer, R., Kirkby, J., and Winkler, P. M.: Enhanced growth rate of atmospheric particles from sulfuric acid, ACP, 20, 7359-7372, 10.5194/acp-20-7359-2020, 2020.

Yao, L., Garmash, O., Bianchi, F., Zheng, J., Yan, C., Kontkanen, J., Junninen, H., Mazon, S. B., Ehn, M., Paasonen, P., Sipilä, M., Wang, M., Wang, X., Xiao, S., Chen, H., Lu, Y., Zhang, B., Wang, D., Fu, Q., Geng, F., Li, L., Wang, H., Qiao, L., Yang, X., Chen, J., Kerminen, V.-M., Petäjä, T., Worsnop, D. R., Kulmala, M., and Wang, L.: Atmospheric new particle formation from sulfuric acid and amines in a Chinese megacity, Science, 361, 278-281, 10.1126/science.aao4839, 2018.

---

## Author Comment (AC2)

Response to Anonymous Referee #2:

*The manuscript of Zhang et al. provides significant measurements of HIOx and their findings show the relation between new particle formation (NPF) and iodine oxoacids, in two polluted urban areas in China. Based on back trajectory analysis, it was found that iodine species mainly originate from marine and terrestrial sources instead of local human activity. Moreover, they calculated the contribution of HIO3 and H2SO4 to growth rate and survival probability, when an NPF event occurred at both examined sites. Their findings indicate that during a NPF day HIO3 promotes the survival of particles with diameters below 3 nm, while no impact is observed for particles between 3 and 7 nm. This study shines light into the crucial role of HIO3 to NPF events under polluted conditions, using experimental measurements.*

*The manuscript is well written and interesting, with an added value of the presented results being from an area of the globe with significant population growth. However, there are several details missing and more thorough discussion should be made in specific sections. Other than that the paper can be recommended for publication after addressing the issues listed below.*

Thank you for your positive feedback and valuable recommendations. We will reply your concerns point-by-point below, incorporating the modifications we made to the revised version of the manuscript. Comments are presented in *blue italic text* followed by our corresponding responses. Any modifications made to the revised manuscript will be highlighted and shown as "quoted underlined text" within our responses.

**General revision:**

We have renumbered all the supplementary figures according to their orders of appearance in the main text.

*1) L155-156: The authors consider that an undefined event is regarded as non-NPF event. Please comment. Also, do the authors believe that no freshly nucleated particles at the size below 3 nm is observed?*

**Response:** Thank you for the comment. As suggested, there are three essential criteria for identifying Nucleation Particle Formation (NPF) events (Kulmala et al., 2012). If all three criteria are met, a day is categorized as an NPF event. Conversely, if none of these criteria are satisfied, the day is classified as a non-NPF event. For days where only some, but not all of these criteria are met, they are designated as undefined events. In certain studies (Deng et al., 2020a; Deng et al., 2020b), it has been suggested that maintaining an undefined category can be advantageous when comparing the characteristics of NPF and non-NPF events. This is because the events during undefined days

exhibit such diversity that it is not appropriate to categorize them as either NPF or non-NPF. However, for the purposes of our study, we determined to classify all undefined days as non-NPF events. This is because of the objectives of our research, which focuses solely on assessing the effect of $HIO_3$ on NPF events. Therefore, it is not necessary for us to further clarify these undefined events.

In fact, it is uncertain whether freshly nucleated particles with diameters below 3 nm are observed on undefined days. The three required features for NPF events are: an elevated concentration of nucleation particles, the persistence of nucleation formation for several hours, and substantial particle growth over several hours. As mentioned earlier, even if sub-3 nm freshly nucleated particles are observed on a day while the nucleation does not persist or there is no significant particle growth, that day was still considered undefined. Conversely, if there is only growth in the size of larger particles without an increase in the concentration of sub-3 nm particles on a given day, we classified it as an undefined event.

*2) L302: Authors mention that the $H_2SO_4$ is lower in cold seasons. Could you elaborate on this?*

**Response:** Thank you for your comment. Sulfuric acid ($H_2SO_4$) primarily forms as a result of the reaction between $SO_2$ and OH in the daytime, and it is mostly removed through condensation onto particles. During the cold season (winter), reduced solar radiation leads to lower OH concentration, thus lower $H_2SO_4$. Additionally, the elevated levels of particle pollution in winter increase the condensation sink (CS) of $H_2SO_4$ on particle surfaces, further reducing its concentration. Consequently, the results indicate a lower concentration of $H_2SO_4$ during the cold season.

*3) L305: Please make a discussion about the calibration issues on HIOx measurements. What is the kind of calibration and the frequency?*

**Response:** Thanks for the suggestions. We have now included a brief discussion of the calibration processes, the references and the expected systematic error of our calibration method. And the manuscript has been revised as below,

[Line 134-145] "$H_2SO_4$ calibration was conducted using a standardized method (Kurten et al., 2012; He et al., 2023). In a nutshell, the calibration of $H_2SO_4$ involved the reaction of an excessive amount of sulfur dioxide ($SO_2$) with a known quantity of hydroxyl (OH) radicals generated by a portable mercury lamp. This mercury lamp is equipped with a filter to intercept the sample air containing water, which, in turn, is photolyzed to produce OH radicals. The convection-diffusion-reaction processes within the chemical ionization inlet can be accurately simulated using a two-dimensional model (e.g., the MARFORCE-Flowtube model) (He et al., 2023), allowing for the quantification of $H_2SO_4$ concentration at the mass spectrometer's entrance. The quantification of the

measured signals for $H_2SO_4$ monomer at both sites are seasonal calibrated with diffusion losses in tube into consideration. Since both $H_2SO_4$ and $HIO_x$ are detected at the collision limit, they share the same calibration factor (He et al., 2021; He et al., 2023). The general systematic error for the detection of $H_2SO_4$ and $HIO_x$ is expected to be within 50% to 200% for field observations (Liu et al., 2021)"

*4) L311: "The results indicate that the $H_2SO_4$ concentrations are generally higher than that of iodine oxoacids at both sites." -> Overall?? Because the time period of the measurements was different at both examined sites. What is the percentage difference among the $H_2SO_4$ iodine oxoacids concentrations at both sites? Why $H_2SO_4$ is constantly higher from iodine oxoacids. Please comment.*

**Response:** Thank you for the comments. While the measurement timeframes in Beijing and Nanjing differ, it is important to note that $H_2SO_4$ concentrations consistently exceeded those of $HIO_3$ throughout all four seasons. This indicates a persistent pattern of higher $H_2SO_4$ concentrations compared to $HIO_x$ over the entire measurement period at both sites.

We computed the ratio of average $HIO_3$ to $H_2SO_4$ concentrations, yielding approximately 10% in Beijing and 9% in Nanjing. The difference in these percentages between the two sites is negligible. And we specify the value in the revised manuscript in **Line [337-338]** as "The ratios of $HIO_3$ to $H_2SO_4$ are consistent at both sites, with about 10% in Beijing and 9% in Nanjing, respectively.".

The consistently higher $H_2SO_4$ concentration is likely attributed to the presence of more abundant precursors, such as $SO_2$, in these anthropogenic inland cities when compared to iodine precursors.

*5) L155-156: The authors support that $H_2SO_4$ is the main contributor to NPF, however in Figure 1 increased $H_2SO_4$ concentrations are related to low NPF frequency. Please provide an explanation of why this feature does not occur here.*

**Response:** Thanks for your comments. We support that $H_2SO_4$, being the primary contributor to NPF, could result in higher nucleation intensity when $H_2SO_4$ levels are elevated. However, it is important to note that there are more factors influencing NPF, such as condensation sink, temperature and concentrations of base molecules. The reason is likely related to the strong temperature dependence of particle nucleation. The high SA exists in summer seasons where the high ambient temperature reduces NPF frequency. This explains why the results depicted in Figure 1 do not show an increase in NPF frequency corresponding to higher $H_2SO_4$ concentrations.

*6) L356-357: "H₂SO₄ concentrations are higher in the spring and autumn, lower in the summer, and lowest in the winter." -> Please add an explanation regarding this seasonal diversity.*

**Response:** Thanks for your comments and good suggestions. $H_2SO_4$ concentration is determined by both the source and the sink. During winter, the lowest OH concentration, which is attributed to weak solar radiation, coupled with the predominant sink caused by the severe particle pollution contributes to the lowest levels of $H_2SO_4$. In summer, while higher OH concentration favors $H_2SO_4$ formation, decreased $SO_2$ concentration from combustion may limit the overall $H_2SO_4$ concentration.

In the revised manuscript, we added the explanation "This variation of $H_2SO_4$ in different seasons is determined by both its source and sink. Winter is characterized by the weakest solar radiation and the heaviest particle pollution, which collectively result in lowest $H_2SO_4$ levels, whereas lower $SO_2$ concentrations in summer could limit $H_2SO_4$ formation." in **[Line 384-387]**.

*7) L400: "The authors found that ambient level of O₃ was not the limiting factor for HIO₃ formation." -> How the authors came to this conclusion? Could you elaborate on this? A reference is needed here.*

**Response:** Thanks for your comments. The authors in the mentioned study conducted tests to assess the sensitivity of $HIO_3$ formation to $O_3$ concentration under 263K (Finkenzeller et al., 2023). If $O_3$ were the limiting factor for $HIO_3$ formation, $HIO_3$ would only begin to form once the $O_3$ concentration surpasses a critical value and would subsequently increase with higher $O_3$ levels. However, the results, as depicted in the following figure (copyright (Finkenzeller et al., 2023)), indicate that there is no significant variation in $HIO_3$ levels with increasing $O_3$ concentration. However, we would like to remark that the sensitivity test conducted by Finkenzeller et al. was intended to reproduce the chamber measurements and to further investigate the rate order for $HIO_3$ reactions. Therefore, $O_3$ not being the rate-limiting factor for $HIO_3$ formation does not mean that $O_3$ is not involved in the formation of its key precursors. Another CLOUD chamber study conducted earlier showed that the injection of $O_3$ can induce effective production of $HIO_3$, and in this study, we have not been able to discuss about the specific influence of $O_3$ on the production of $HIO_3$ due to the limited understandings and direct measurements of $HIO_3$ precursors at both sites. To avoid confusion in the main text, we have removed the related statement and add some discussions about the relationship between $O_3$ and $HIO_3$ at both sites. Therefore, the expression in original manuscript **[Line 400-408]** has been revised as following in **[Line 434-444]**.

**[Line 434-444]** "The diurnal patterns show that the maximum of daytime $HIO_3$ concentration mimic that of $O_3$ in all seasons, indicating that $O_3$ may influence terrestrial $HIO_3$ formation. However, the

role of $O_3$ in $HIO_3$ formation can be multifaceted and warrants more thorough discussion in the future studies with extensive measurements of other iodine compounds in inland regions, especially urbanized areas. On one hand, chamber studies have shown direct involvement of $O_3$ in the formation of $HIO_3$ precursors in less chemically complex scenarios. On the other hand, $O_3$ has been proved to stimulate the release of iodine compounds in marine environments from surface sea water (Carpenter et al., 2013) and similar processes are likely to occur in urban environments as well. Additionally, temperature may also have favourable impacts on both the formation of $HIO_3$ and the release of iodine precursors, which will be discussed in the next section."

[Figure]

**Figure R1.: Sensitivity of $HIO_3$ to changes in $O_3$ concentrations under the assumption of different hypothesised mechanisms, and comparison with observations at the CLOUD chamber (copyright from Supplementary Fig. 1 in (Finkenzeller et al., 2023)).**

*8) L401: "...indicating that $O_3$ may influence terrestrial $HIO_3$ formation." -> It would be useful to provide a scatter plot for $O_3$ vs. $HIO_3$ to advocate this conclusion.*

**Response:** Thanks for your comments and good suggestions. As demonstrated in Fig. R2 for the entire observation period at both sites and in Figure 4 for the distinct four seasons, there is a positive correlation between $HIO_3$ concentration and $O_3$ levels. This suggests the potential influence of $O_3$ on the measured concentrations of $HIO_3$. However, the exact role of $O_3$ on the formation of HIO3 warrants deeper investigation, especially in chemically complex environments (like urbanized areas in this study) with limited detection of iodine species. On one hand, as depicted in Finkenzeller et al., $O_3$ can directly participate in the formation pathways of $HIO_3$ with critical precursors such as IOIO (Finkenzeller et al., 2023). On the other hand, $O_3$ has been proved to stimulate the liberation of volatile iodine species from ocean surfaces, indicating that $O_3$ can be related with the emission of $HIO_3$ precursors as well. It should also be noted that the positive correlation between $HIO_3$ concentration and ozone in this study is more pronounced in cold months at both sites and the correlations become weaker in warm conditions. The reason could be the stabilized emissions of

iodine sources under warm conditions, where increasing ozone will no longer promote the production of HIO₃. Moreover, $O_3$ levels are higher in summer at both sites with simultaneous higher temperatures, resulting in more active emission of volatile iodine species, which could further contribute to iodic acid formation. The relationship between ozone and HIO₃ is multifaceted, and we have restated the role of ozone as a reactant with HIO₃ precursors as well as a potential factor influencing the release of iodine in inland regions as shown in the previous response to comment #7.

[Figure]

**Figure R2. The correlation between $O_3$ and HIO₃ at (a) BUCT/AHL and (b) SORPES. All points are daytime values during the measurements and are color-coded with temperature. The median values of different seasons at both sites are shown in the color bar axis.**

*9) L403: "This mechanism mainly explains…" -> In which exactly process the authors refer to? Please explain.*

**Response:** Thanks for your comments. We refer to the $O_3$-stimulated emissions of inorganic iodine species from the ocean surface, which is a critical process for the formation of $I_2$ and HOI through the reaction of iodide with $O_3$. Our mention of this here is intended to propose a potential similar influence of $O_3$ on HIO₃, specifically regarding the effect of $O_3$ on the precursor of terrestrial HIO₃.

*10) L406: "Another possibility is that air temperature may strongly perturb the formation of HIO₃ …" -> A reference is needed here. This means that the augmentation of temperature involves increased HIO₃ concentrations? I would be careful of using the verb "perturb" here.*

**Commented [XH1]:** This should be modified. We had discussed that we cannot explicitly explain the correlation between ozone and HIO3. Please say O3 is also higher in the summer and it could be related to higher temperature, therefore higher emission. Another possibility is that higher O3 introduced higher iodine emission. In any case, please emphasize that Finkenzeller is talking about chemistry production while here is might more be related to sources.

**Commented [XH2R1]:** Oh, I see you explain these in the next questions. Anyway, refer to these replies below. But do separate the concept of chemical production and sources.

**Commented [u3R1]:** Will this refined version be more clear?

**Response:** Thanks for your comments. Our speculation regarding the influence of temperature on HIO$_3$ formation is based on the observation of a similar diurnal pattern between HIO$_3$ and temperature. There are two key aspects of the temperature effect on HIO$_3$ concentration. First of all, higher T may lead to increased emission of HIO$_3$ precursors, such as CH$_3$I, which we have discussed in 3.2.3 in the manuscript. Secondly, the HIO$_3$ formation reactions are temperature-dependent and could be promoted under warmer conditions, and we have incorporated relevant reference (Finkenzeller et al., 2023) in the revised manuscript. And in the revised manuscript, we have replaced "strongly perturb" with "have a favorable influence on both".

*11) L428-429: "...land indicates that the continental outflows may also play a significant role in transporting HIO$_3$ precursors." -> The authors consider that air masses coming from South have continental properties. However, in Figure S4a the southern cluster can travel over the sea, showing marine properties. Please comment on this?*

**Response:** Thank you for your comment. The southern cluster could travel both over the sea and the inland, whereas the northern wind always comes from the continental areas. Therefore, the northern cluster instead of the southern one would indicate the continental outflow better and we corrected the expression in manuscript as following.

**[Line 465-466]** "Additionally, the air mass travels from northern wind may also carry substantial precursors of HIO$_3$, indicating the potential terrestrial sources of iodine."

*12) L434: "...impact from residential coal burning and fossil fuel combustion power plant in Beijing is the largest." -> Are there any BC measurements in AHL/ BUCT station? A figure for BC vs. HIO$_3$ would be useful to advocates the author's conclusion.*

**Response:** Thank you for your valuable suggestion. To enhance our points, we've included Fig. R3 according to your advice, which elucidates the relationship between HIO$_3$ and BC. The findings demonstrate a notable increase in HIO$_3$ concentration during periods of low BC, further supporting our conclusion.

In the revised manuscript, we added the sentences "The negative correlation between HIO$_3$ and BC shown in Fig. S6 further demonstrate the irrelevance of winter pollution on the HIO$_3$ in Beijing." in **[Line 472-474].**

[Figure]

**Fig R3. HIO₃ concentration in different BC level bins.**

*13) L559: "Only events with clear growth…" -> This assumption means NPF events of class I. However, the classification of NPF episodes is not easily identifiable from the text. A classification could be added to the main text (2.2.1) and in the respective Table.*

**Response:** Thank you for your good suggestions. We complete the classification method for NPF events in main text as following sentences and update the Table S1 for specific NPF events A and B.

**[Line 169-170]**: "Furthermore, NPF events exhibiting obvious nucleation and clear growth of fresh nucleation particles were categorized as "NPF-A," while the remaining NPF events were designated as "NPF-B"."

*14) L567: "…environments with higher CoagS, such as Beijing, a fixed GR enhancement can lead to larger SP…" -> What does fixed GR enhancement mean? Please clarify.*

**Response:** Our apologies for not being clear. In this section, we were trying to discuss about the sensitivity of SP enhancement against GR enhancement, which is controlled by CoagS.

For Eq. (8), by considering that $d_{p1}$, $d_{p2}$ and CoagS are constant, we can further combine the constant values besides CoagS as a new term A.

Where

$$A = \frac{d_{p1}}{m-1}\left[\left(\frac{d_{p2}}{d_{p1}}\right)^{1-m} - 1\right],$$ (R1)

and Eq. (8) can be simplified as

$$SP = \exp\left(\frac{CoagS}{A} \cdot \frac{1}{GR}\right).$$ (R2)

By assuming m=1.7, it can be deduced that A<0, and if we take the derivative of SP with respect to GR for Eq. (R2), we further get the following Eq. (R3):

$$\frac{\Delta SP}{\Delta GR} = -\frac{1}{GR^2} \cdot \frac{CoagS}{A} \cdot SP.$$ (R3)

For a given GR value and GR enhancement (ΔGR), the enhancement of SP (ΔSP) is subject to the value of CoagS. The larger the CoagS is, the larger enhancement of SP can be expected.

For the two cases studied in this section, the GR values are similar while on Jun 21, 2021, the CoagS is 5 times larger than May 29, 2021, leading to a more pronounced enhancement of SP on the former day. The expressions we were using was somehow confusing, and we have revised the statement as below,

**[Line 608-609]** "This suggests that in polluted environments with higher CoagS, such as Beijing, SP enhancement can be more sensitive to GR enhancement."

*15) L568-569: "Results in both Fig. 7 and Fig. S5 show that the median contribution of HIO$_3$ to the GR of particles in the 1.5-3 nm size range is 7.4% using the MOD method…" -> This contribution is referring to all NPF episodes in Beijing?*

**Response:** Thanks for your comment. We apologize for any confusion caused by our previous statement. As previously mentioned, our study specifically concentrated on NPF-A events when considering the contribution of HIO$_3$ to GR and SP enhancement. Therefore, when we refer to this contribution (7.4%), it pertains exclusively to all the NPF-A events in Beijing.

*16) L569-571: "…whereas the contribution is only around 3% and 2% using the APT-x and APT-y methods, respectively. This is resulted from the difference in the*

*measured GR calculated using either the APT or the MOD methods." -> Measured or calculated GR?? Therefore, this variation derives from the different estimated GR. What is the amount of uncertainty?*

**Response:** Thanks for your good comment. We apologize for any lack of clarity in our earlier manuscript. It's crucial to emphasize that both the APT and MOD methods stem from the measured particle number size distribution (PNSD), signifying that these values are derived through calculations based on the measured results rather than entirely theoretical predictions. Furthermore, in order to elucidate the distinctions among these methodologies, we have included additional explanations in the revised supplementary information in **[Line 1296-1317]**.

S5. The calculation method of GRs in NPF events

The growth of newly formed particles for NPF events are often reflected by the collective shift of measured particle size distribution towards larger sizes as time evolves, and it is unfeasible to track the growth of a single particle based on the measurements. Therefore, both approaches used in this study (mode-fitting and appearance time) are referred to as collective approaches and the GRs in this study are the estimated ones (Stolzenburg et al., 2023). While using mode-fitting method, the growth trajectory of new particles is represented by the peak diameter ($d_p$) of the nucleation mode, which is determined after applying log-normal distributions to the measured size distribution (Kulmala et al., 2012). For measured particle number size distribution at each time ($t$), there will be a $d_p$, and the value of GR (GR$_{mode}$) is derived by a linear fit to the $d_p$ vs $t$.

Instead of tracking the shift of peak diameter for a given time period, appearance time method seeks to find the time it takes ($\Delta t$) for the particle to grow between instrument size bins ($\Delta d_p$). In this study, we take the time ($t$) that the measured concentration of particles reaches its half maximum for each $d_p$. For each particle size bin ($d_p$) of the instrument used at BUCT/AHL, there will be a $t$, and the value of GR (GR$_{apt}$) is derived by a linear fit to the $d_p$ vs $t$. Additionally, we believe it would be conceptually more correct to consider diameter as the independent variable when fitting the GRs using appearance time method although previous studies commonly take appearance time as the independent variable and diameter as the dependent variable when fitting the GRs. that is because each data point corresponds to a precise size bin, and any variation (largely stemming from uncertainty due to atmospheric heterogeneity) among the data points primarily exists in appearance time. Consequently, the fitting method with diameter as the independent variable was named as APT-y and the widely used one with time as the independent variable was called as APT-x.

However, it's important to note that the ongoing debate regarding the accuracy between the MOD and APT methods remains unresolved. Hence, this variation is derived from the different estimated GRs. For instance, the GRs for sub-3 nm particles determined using APT-x, APT-y, and MODE on May 26, 2021, were 3.17, 5.13, and 2.22 nm h$^{-1}$, respectively, resulting in an average discrepancy of 1.93 nm h$^{-1}$ between both APT methods and MODE. However, these values changed to 1.50, 1.92, and 1.93 nm h$^{-1}$, respectively, on Jun 21, 2021, leading to a comparatively minor deviation. Moreover, the disparity in GRs for 3-7 nm particles across these methods is more

substantial as expected, given that the GRs for these particles are consistently more than those of sub-3 nm particles. The result reveals an average discrepancy among those methods of 1.5 nm h$^{-1}$ for sub-3 nm particles and 3.1 nm h$^{-1}$ for 3-7 nm particles, reflecting the level of uncertainty associated with GR determination.

**Technical corrections:**

*L24: "New particle formation processes contribute" -> New particle formation contributes*

**Response:** Thank you for good suggestion. We have updated "New particle formation contributes significantly to the number concentration of ultrafine particles (UFP, d≤100nm), and have great impacts on human health and global climate." in the lines 24-25 in the revised manuscript

*L25: "··· ultrafine particles (UFP), and have great···" -> ultrafine particles (UFP; d≤100 nm)*

**Response:** Thank you for good suggestion. We added the diameter range according to your advice.

*L27: "...proved to dominate NPF events at some sites." -> proved to dominate NPF.*

**Response:** Thank you for good suggestion. Deleted the "events".

*L28: "... we conducted a long-term comprehensive observation of gaseous…" -> ... long-term measurements of gaseous ...*

**Response:** Thank you. Corrected.

*L30: "... concentration in urban…" -> ... concentration in both urban….*

**Response:** Thank you. Added "both".

*L31: "HIO₃ concentration is…" -> HIO₃ is ….*

**Response:** Thank you for good suggestion. Deleted the "concentration".

*L32-33: " ...and is lowest in winter…" -> ... and is lowest in winter by xxx% and xxx%, respectively. HIO₃ exhibits more prominent variation than H₂SO₄ in both urban sites.*

**Response:** Thank you for good suggestion. We have updated "and is lowest in winter by 96% and 75%, respectively. HIO₃ exhibits more prominent variation than H₂SO₄ in both urban sites." in the **[lines 31-32]** in the revised manuscript

*L34: "...temperature, radiation and ozone…" -> ...temperature, solar radiation and ozone…*

**Response:** Thank you. Added "solar".

*L40: "...suggesting HIOx are non-negligible contributor to UFPs in polluted urban areas." -> ...suggesting that HIOx are significant contributor to UFPs in polluted urban areas.*

**Response:** Thank you. Corrected.

*L46-49: "Primary aerosol emissions include natural sources including the emission of sea spray, release of soil mineral dust, emission of biomass burning smoke, and the injection of volcanic debris (Claudio Tomasi, 2017) and anthropogenic emissions such as fuel combustion, industrial processes and transportation (Claudio Tomasi, 2017)." -> Primary aerosol emissions stem from natural sources, including sea spray, soil mineral dust, biomass burning, and volcanic debris (Claudio Tomasi, 2017), and anthropogenic sources such as fuel combustion, industrial processes and transportation (Claudio Tomasi, 2017).*

**Response:** Thank you. Corrected.

*L53: "...form new particles under atmospheric conditions..." -> ... form new particles under appropriate atmospheric conditions...*

**Response:** Thank you. Added "appropriate".

*L57: "...influence cloud formation and have climatic effects (Kerminen et al., 2005)." -> Use a more recent reference, e.g. Kalkavouras et al. (2019); Jiang et al. (2021)*

**Response:** Thank you. We have added new references as you commended (Jiang et al., 2021; Kalkavouras et al., 2019).

*L59: "···These small particles (< 50 nm) can penetrate···" -> You mean UFP, thus use ≤100 nm instead of 50 nm in the parenthesis.*

**Response:** Thank you. Corrected.

*L60: "...understanding NPF processes is…" -> delete the word "processes"*

**Response:** Thank you for good suggestion. Deleted.

*L61: "...terms of predicting climate change..." -> Use evaluating instead of predicting*

**Response:** Thank you. Corrected.

*L70: "…vapours depending on the particle sizes. In urban Beijing,…" -> …vapours depending on the particle size. In Beijing…*

**Response:** Thank you for good suggestion. Deleted the "urban".

*L76-77: Use capital letter for O' Dowd.*

**Response:** Thank you. Corrected.

*L85: "…that HIO$_3$ (with HIO$_2$)…" -> Please clarify the content in the parenthesis. Its vague*

**Response:** Thank you for good comment. The phrase "with HIO$_2$" in the parenthesis means the critical role of HIO$_2$ in stabilizing HIO$_3$ clusters. Therefore, we corrected it as "(stabilized by HIO$_2$)".

*L98: "…in urban Beijing from 2019 to 2021,…" -> …in urban Beijing from January 2019 to October 2021, …*

**Response:** Thank you. Corrected.

*L107-108: "The measurement in urban Beijing was conducted from January 2019 to October 2021. The site locates on the fifth floor…" -> Measurements in urban Beijing were conducted from January 2019 to October 2021, on the fifth floor…*

**Response:** Thank you. Corrected.

*L107-115: It would be useful to provide a figure with the exact location of both stations in main text or in the SM.*

**Response:** Thanks for the suggestion. We have added a figure showing the location of two measurement sites in the supplementary material (Fig. S1), as shown in the following.

[Figure]

**Figure R4: The geophysical distribution of two measurement sites (BUCT/AHL in Beijing, China and SORPES in Nanjing, China) of this study.**

*L122 and L124: Be sure about the unit of measurement*

**Response:** Thank you for your good comments. The second unit of measurement has been corrected as "Th Th$^{-1}$".

*L125: "...from January 2020 to February 2020." -> or ...from March 2020 to February 2020. ?*

**Response:** Thank you. In fact, we used LToF from March 2019 to December 2019 and HToF from January 2020 to February 2020 at SORPES in Nanjing.

*L135-142: Long sentence, please rephrase*

**Response:** Thank you for your good suggestions. We have rewritten this part in the revised manuscript as following.

**[Line 148-154]**: The particle number size distribution (PNSD) from approximately 1 nm to 10 μm at the AHL/BUCT station was measured. This was done using a diethylene glycol scanning mobility particle spectrometer (DEG-SMPS, 1-4.5 nm) (Jiang et al., 2011), equipped with a miniature cylindrical differential mobility analyzer (mini-cy DMA) (Cai et al., 2017a). In addition, we utilized a homemade particle size distribution system (PSD, 3 nm-10 μm) (Liu et al., 2016). At the SORPES station, the PNSD was measured using an Aerodynamic Particle Sizer (APS, TSI, APS-3321, USA, 500-1000 nm) and two SMPSs equipped with a TSI long-DMA (TSI Inc., model 3081) and a TSI nano-DMA (TSI Inc., model 3085).

*L168: "...correction factor for mass flux as a function..." -> ... correction factor for mass flux (Fuchs and Sutugin, 1970) as a function...*

**Response:** Thank you for your good suggestions. Added the reference.

*L174: "To quantify if notable growth is to occur, especially at sizes below a few nanometers, it is…" -> What do you mean a few nanometers? Please explain.*

**Response:** Thanks for your comment. We apologize for the previous unclear expression. We tended to emphasis the significance of coagulation sink for newly formed nucleation particles, which plays a great role in quantifying the occurrence of growth. Here is the revised version:

**[Line 190-191]:** To quantify if notable growth is to occur, especially for sub-3 nm particles, it is crucial to understand the loss process of fresh particles.

*L225: "The counterbalance of CoagS and GR considerably affects the survival of small clusters." -> Please add a reference here.*

**Response:** Thank you for your good suggestions. Added the reference (Mahfouz and Donahue, 2021).

*L269: Put the sentence in line 269, after the equation (11).*

**Response:** Thanks for your comment. Corrected.

*L286: "Based on current knowledge about $HIO_3$ formation pathways,…" -> Please add a reference here.*

**Response:** Thanks for your comment. Added (Finkenzeller et al., 2023).

*L315: "...from none to more than three quarters of the days…" -> Why present it as ¾ and not e.g. 75% ?*

**Response:** Thanks for your suggestions. In revised manuscript, "75%" instead of the "three quarters".

*L347: "To better understand the roles of the studied acids in new particle formation and growth, we further…" -> To better understand the roles of the studied acids in NPF, we further….*

**Response:** Thanks. Corrected.

*L358: "The $HIO_3$ concentrations measured at the two sites are significantly lower*

*than that at pristine coastal…" -> How low? Please add a percentage to express the variation of HIO$_3$.*

**Response:** Thanks for your suggestions. The HIO$_3$ concentrations at pristine coastal during the NPF events reached $10^8$ cm$^{-3}$, whereas the maximums observed at Beijing and Nanjing were only around $10^6$ cm$^{-3}$. Thus, we concluded that "The HIO$_3$ concentrations measured at the two sites are significantly lower, approximately two orders of magnitude less than that at pristine coastal site"

*L363: "... concentrations in Beijing and Nanjing are comparable to that in Helsinki, Finland." -> A reference is needed here.*

**Response:** Thanks for your suggestion. Reference was added in new manuscript.

*L387-388: "The distinct diurnal variation in HIO$_x$ concentration with around one order of magnitude implies fast in-situ chemistry." -> This clear diurnal variation is observed during summer? It's vague. Please clarify.*

**Response:** Thank you for good suggestions. We clarified it as following "As depicted in Fig. 2, this pronounced diurnal variation in HIO$_x$ levels is consistently observed at both sites throughout the entire measurement period, regardless of the season. The peak concentrations of HIO$_x$ consistently exceed the minimum levels by a factor of approximately one order, with this difference being particularly significant, especially for HIO$_3$ during summer. The distinct variation from day to night suggests that HIO$_x$ formation is primarily occurring in situ, rather than being transported from other regions." in **[Line 418-423]** of manuscript.

*L390: "This phenomenon is pronounced during summer daytime when daily maximum of H$_2$SO$_4$ appears around 2 hours earlier than that of HIO$_3$" -> This behavior was also apparent in autumn. It would be nice to refer this with some comments.*

**Response:** Thank you for pointing this out. After checking the diurnal data and Figure 2, we realize that the delay of HIO$_3$ daytime peak compared to H$_2$SO$_4$ is a common phenomenon regardless of season variation. Therefore, we revise the statement to avoid the specific discussion of the seasonal pattern of this phenomenon:

**[Line 424-426]** "This phenomenon is pronounced regardless of season at both sites with the daily maximum of H$_2$SO$_4$ appears around 1-2 hours earlier than that of HIO$_3$."

*L393: "...cycle of H$_2$SO$_4$ follow that of radiation in spring, summer and winter." -> But not for fall? From Fig. 2 it is clear that the daily pattern of H$_2$SO$_4$ follows that of solar radiation.*

**Response:** Thank you for good suggestions. We apologize for the error in the wording and have removed the restriction related to seasons in the revised manuscript. At both sites, the diurnal

variation of H₂SO₄ track the solar radiation very well. However, specially at SORPES, the summertime peak for H₂SO₄ track the diurnal variation of SO₂ as well, indicating the influence of regional emission control.

We have revised the statement as below,

[Line 428-431] "In summer, however, owing to effectiveness of long-term emission reduction, SO₂ concentrations can be low enough to limit the production of H₂SO₄ at SORPES (Ding et al., 2019), so the daytime peaks of H₂SO₄ tend to occur when SO₂ reached its daily maximum (Yang et al., 2021)."

*L429-430: "...and terrestrial precursors (Li et al., 2014; Wang et al., 2017) may…" -> Terrestrial precursors such as?*

**Response:** Thank you for good suggestions. We revised as "terrestrial precursors such as soil fumigants".

*L472: "...terrestrial biomes (Sive et al., 2007), and minor wetlands (Dimmer et al., 2001), biomass burning…" -> ...terrestrial biomes (Sive et al., 2007), minor wetlands (Dimmer et al., 2001), and biomass burning…*

**Response:** Thank you. Corrected.

*L525-526: "When P is below 50 in clean environment or 100 in polluted urban cities, the SP of the sub-3 nm particles is agreed with the atmospheric observations." -> A reference is needed here.*

**Response:** Thank you for your good suggestions. We again added the reference (Kulmala et al., 2017) where the P was proposed and listed for observations at various sites.

*L526: "As shown in Fig. S6, the…" -> The Fig. S6 in which station is referred to? The right y axis expresses the P? It is not clear.*

**Response:** Thank you for your comments. The Fig. S8 (previously Fig. S6) is referred to in Beijing and the right y axis actually represents 'P'. We have replaced the unclear graph with a new one.

*L535-537: "Under the typical CoagS (around 0.0025 s⁻¹) at both sites, the SP could be enhanced by more than two orders of magnitude when GR is varied from 1 to 10 nm h⁻¹." -> It would be useful to provide a new figure, showing this CoagS value, because it's difficult for a reader to find/ see the value of 0.0025 s⁻¹.*

**Response:** Thank you for your good suggestions. In the revised manuscript, we have included

Figure R5 to illustrate the sensitivity of the GR on the SP while keeping the CoagS at a constant value 0.0025s⁻¹.

[Figure]

**Figure R5. The particle survival probabilities for particles in the size range of 1.5-3 nm and 3-7 nm as a function of the GR when the CoagS was set at 0.0025 s⁻¹.**

*L537-538: "Increased GR caused by additional condensing vapours enables faster growth, which in turn facilitates the survival of sub-10 nm particles from coagulation scavenging." -> Please add a reference here.*

**Response:** Thanks for your good suggestions. We added a reference (Kuang et al., 2012) that proposed an enhanced factor for GR from vapors other than $H_2SO_4$ and highlighted its impact on the SP.

*L540: "A limited variation of GR was suggested to cause considerable variation of SP (Cai et al., 2021a)." -> Repetition. The same sentence is written in the beginning of paragraph (line: 531-532).*

**Response:** Thank you. We deleted the repeated expression.

*L562-563: "The results in Table S2-S4 show that the contributions of HIO₃ to GR on May 25 and 26 were lower than 5%, whereas the contribution was more than 10% on May 29" -> In Table S4 the contribution of HIO₃ to GR for <3 nm particles is 9.4% (i.e. below 10%) on May 29. Furthermore, in this paragraph (L: 555-574) the authors analyze the MOD method. Why they add Tables S2 and S3?*

**Response:** Thank you. We apologize for the lack of rigor. We corrected it as following in manuscript.

**[Line 604-605]**: "The results in Table S2-S4 show that the contributions of HIO₃ to GR on May 25 and 26 were lower than 5%, whereas the contribution was about 10% on May 29."

With respect to the Tables S2 and S3, we decided to display them in order to minimize potential systematic uncertainties arising from the GR calculations. To the best of our knowledge, it is not possible to definitively determine which of the MOD, APT-x, and APT-y methods is the most accurate.

*L598-599: "At the SORPES station from June to November 2019, HIO₃ contributes 6.1% (median) and 6.7% (mean) to sub-3 nm particles growth compared to H₂SO₄." -> You mean that HIO₃ exhibits higher contributions compared to H₂SO₄ ?*

**Response:** We apologize for the confusion in the statement. The percentage here refers to the fact that the contribution of HIO₃ to sub-3 nm particle growth accounts for 6.1% (median) and 6.7% (mean) of H₂SO₄.

And we have revised the statement as below,

**[Line 638-640]**: "At the SORPES station from June to November 2019, the contribution of HIO₃ to sub-3 nm particle growth accounts for 6.1% (median) and 6.7% (mean) of H₂SO₄."

*Fig. S9: In the caption SORPES instead of SOREPS.*

**Response:** Thank you. Corrected.

**References**

[revised manuscript text omitted]